# Enhancer Regulation of Dopaminergic Neurochemical Transmission in the Striatum

**DOI:** 10.3390/ijms23158543

**Published:** 2022-08-01

**Authors:** Laszlo G. Harsing, Joseph Knoll, Ildiko Miklya

**Affiliations:** Department of Pharmacology and Pharmacotherapy, Semmelweis University, Nagyvarad ter 4, 1089 Budapest, Hungary; harsing.laszlo@med.semmelweis-univ.hu (L.G.H.J.); knoll.jozsef@med.semmelweis-univ.hu (J.K.)

**Keywords:** (−)BPAP, catecholaminergic activity enhancer effect, trace amine-associated receptor 1, [^3^H]dopamine release, rat striatum

## Abstract

The trace amine-associated receptor 1 (TAAR1) is a Gs protein-coupled, intracellularly located metabotropic receptor. Trace and classic amines, amphetamines, act as agonists on TAAR1; they activate downstream signal transduction influencing neurotransmitter release via intracellular phosphorylation. Our aim was to check the effect of the catecholaminergic activity enhancer compound ((−)BPAP, (*R*)-(−)-1-(benzofuran-2-yl)-2-propylaminopentane) on neurotransmitter release via the TAAR1 signaling. Rat striatal slices were prepared and the resting and electrical stimulation-evoked [^3^H]dopamine release was measured. The releaser (±)methamphetamine evoked non-vesicular [^3^H]dopamine release in a TAAR1-dependent manner, whereas (−)BPAP potentiated [^3^H]dopamine release with vesicular origin via TAAR1 mediation. (−)BPAP did not induce non-vesicular [^3^H]dopamine release. N-Ethylmaleimide, which inhibits SNARE core complex disassembly, potentiated the stimulatory effect of (−)BPAP on vesicular [^3^H]dopamine release. Subsequent analyses indicated that the dopamine-release stimulatory effect of (−)BPAP was due to an increase in PKC-mediated phosphorylation. We have hypothesized that there are two binding sites present on TAAR1, one for the releaser and one for the enhancer compounds, and they activate different PKC-mediated phosphorylation leading to the evoking of non-vesicular and vesicular dopamine release. (−)BPAP also increased VMAT2 operation enforcing vesicular [^3^H]dopamine accumulation and release. Vesicular dopamine release promoted by TAAR1 evokes activation of D2 dopamine autoreceptor-mediated presynaptic feedback inhibition. In conclusion, TAAR1 possesses a triggering role in both non-vesicular and vesicular dopamine release, and the mechanism of action of (−)BPAP is linked to the activation of TAAR1 and the signal transduction attached.

## 1. Introduction

There has been a major breakthrough in understanding the neurobiology of the central nervous system with the discovery of trace amine-associated receptors (TAARs) [1,2,3]. Later, as the different members of this receptor family were characterized, a large amount of evidence accumulated for the role of TAAR1 in the regulation of catecholaminergic neurotransmission [4].

TAAR1 was found in several brain areas containing dopaminergic, noradrenergic, and serotonergic neuronal cell bodies or axon terminals: the caudate nucleus, putamen, substantia nigra, accumbens nucleus, ventral tegmental area, locus coeruleus, and raphe nuclei of mouse, monkey, and human brain [1,5,6]. It was shown using RT-PCR that TAAR1 can be detected in several human abdominal organs including stomach, liver, pancreas, small intestine, spleen, kidney, prostate gland as well as in lung and skeletal muscles [1,3].

TAAR1 has a cellular location in neurons and is functioning in the intracellular space [2]. TAAR1 is a G protein-coupled receptor and its signaling is regulated by stimulatory Gs protein. The binding of agonist ligands to TAAR activates Gs proteins, and increases adenylate cyclase activity and cAMP production [1,2,7]. Several laboratories expressed TAAR1 in cell lines and demonstrated intracellular cAMP production using different TAAR1 receptor agonist and antagonist compounds [8,9].

It has been shown that TAAR1 and dopamine transporter (DAT) are co-localized in substantia nigra neurons and other brain areas rich in dopaminergic axon terminals [10]. A possible interaction between TAAR1 and DAT is further supported by the fact that many drugs that substrate for DAT also act as agonist on TAAR1. TAAR1 regulates DAT in shifting its bidirectional operation from the uptake mode to a release mode [11]. Decreased activity of the transporter in uptake mode was a protein kinase (PK) A/PKC- mediated phosphorylation process, whereas only PKC phosphorylation mediated the release mode operation of the carrier [12].

There might also be a functional link between TAAR signaling and D2 receptor-mediated presynaptic negative feedback inhibition in dopamine neurons [13]. It has been shown that activation of TAAR1 signaling causes an increased release of newly synthesized dopamine [14], which then leads to activation of dopamine D2 autoreceptor-mediated feedback mechanism and inhibition of dopaminergic neuronal activity [10]. Thus, according to the current view, TAAR1 and its downstream signaling regulate excitability of dopaminergic neurons via interaction with dopamine transporter and dopamine D2 receptor-mediated feedback inhibition [8,9,12].

Two members of the TAAR family, TAAR1 and TAAR4 were identified as receptors for trace amines [15,16]. Phenylethylamine, tyramine, and tryptamine are decarboxylated metabolites of the amino acids phenylalanine, tyrosine, and tryptophan (Figure 1). These amines are present in the brain in unusually low concentrations and were designated as trace amines [17,18,19]. Similar to classical biogenic amines, trace amines bind to TAAR1—their affinities are, however, different: phenylethylamine and tyramine bind to the TAAR1 with high affinity, whereas the binding affinity of octopamine, dopamine, and serotonin to this receptor is much lower [12].

Trace amines are stored in cytoplasmic pools along with the newly synthesized biogenic amines within the presynaptic nerve endings [3]. Their release occurs from reserpine-insensitive stores and it is a non-quantal release that does not depend on membrane depolarization [17]. Similar to classical biogenic amines, trace amines are metabolized by monoamine oxidase (MAO): phenylethylamine is a MAO-B substrate [3,20], tyramine, and other trace amines are metabolized by both A and B isoforms of MAO [12]. It was found that the trace amines interact with membrane monoamine transporters but lack direct interaction with monoamine autoreceptors [21,22].

TAAR1 and TAAR4 were also shown to be sensitive to several other, mostly exogenous compounds, including amphetamine-like psychostimulant drugs and thyroamines [3,19]. Alterations of the chemical structure of the parent molecule phenylethylamine result in a series of compounds with distinct neurochemical profiles. (±)Amphetamine, in which the α carbon atom is substituted with a methyl group in the side chain, is considered a potent catecholamine releaser. (±)Methamphetamine (Figure 1), in which both the α carbon and the nitrogen atoms are substituted with methyl groups, exhibits a similar releasing effect from the cytoplasmic neurotransmitter pool in the nerve endings. The release of biogenic amines elicited by (±)methamphetamine or the trace amines exhibits a number of similarities: this release affects resting outflow, the efflux is external Ca^2+^-independent, and the newly synthesized biogenic amines are primarily involved in the efflux process.

Several analogues of (±)amphetamine or (±)methamphetamine were synthesized, resulting in a series of potent psychostimulant drugs, namely *p*-hydroxyamphetamine, methvlenedioxymethamphetamine, methylphenidate, etc. [3,23]. Phenylethylamine, (±)methamphetamine, N,α-diethylphenethvlamine (DBA), and (−)PPAP represent a series of analogue compounds, which differ in the length of alkyl substituents in the side chain α carbon and nitrogen atoms (Figure 1). DEA, in which both the α carbon and the nitrogen atoms in the side chain are substituted with ethyl groups, does not affect catecholamine efflux, has no psychostimulant effect, and it is frequently used as a pre-workout supplement [24].

Parallel with the discovery of TAARs, Knoll and coworkers [25] described the pharmacological actions of a series of aryl substituted 2-propyl-aminopentanes on neurochemical transmission in the brain [26]. These compounds were capable of increasing the electrical stimulation-induced release of dopamine, noradrenaline, and serotonin in low femto/picomolar concentrations in in vitro conditions without altering the resting release of these biogenic amines [27,28]. Compounds exhibiting these unique pharmacological actions are designated as catecholaminergic activity enhancer (CAE) drugs [25]. The most well-characterized member of the enhancers is (−)BPAP ((*R*)-(−)-l-(benzofuran-2-yl)-2-propylaminopentane hydrochloride) besides PPAP and IPAP (Figure 1). In the case of PPAP, IPAP, and BPAP, the negative enantiomers (R configurations) proved to be biologically more active [29].

Although compounds shown in Figure 1 can be derived from phenylethylamine or other trace amines, there are differences in the ring or the attached alkylamine side chains. A phenyl ring can be found in the structure of (−)PPAP and (±)methamphetamine, whereas there are indole or benzofuran ring in (−)IPAP or (−)BPAP, respectively. The catecholamine enhancer drug (−)PPAP influences primarily the catecholaminergic transmission, (−)IPAP exerts its major effect on serotonergic transmission, whereas (−)BPAP shows approximatively equal potency on both biogenic amine transmissions [26]. We assumed that these differences can be linked to the aryl substitution of the 2-propyl-aminopentane side chain. Compounds ((−)PPAP, (−)IPAP, (−)BPAP) (Figure 1) that contain 2-propyl-aminopentane moiety are selective enhancers of the impulse propagation mediated-release of catecholamines or serotonin, but leave MAO activity practically unchanged [25].

CAE compounds made it possible to analyze the enhancer regulation of the neurochemical transmission in the brain and led to identifying their role in life expectancy [30,31], tumor genesis [32,33], and psychiatric diseases such as depression, mood disorders, and stress-related anxiety [34,35,36]. (−)BPAP was extensively investigated in a number of behavioral and longevity studies, and its antitumor effect has also been demonstrated [31,32]. The potential uses of (−)BPAP in these clinical indications are believed to relate to its CAE effects.

The precise mechanism behind the enhancer activity is not yet fully clarified [37]. Therefore, the aim of the present study was to investigate whether drugs possessing CAE effects increase dopaminergic neurochemical transmission via activation of the TAAR1 signaling pathway. For this purpose, we prepared slices from rat striatum loaded with [^3^H]dopamine and measured the effects of the CAE drugs and TAAR1 ligands on the vesicular and non-vesicular [^3^H]dopamine release. Comparison of the effects of these two groups of compounds led us to the conclusion that the enhancer drugs represent a novel series of TAAR1 agonists. Furthermore, we have also obtained direct evidence that TAAR1 signaling may possess a central role in the regulation of presynaptic dopaminergic neurotransmission.

## 2. Results

### 2.1. Characterization of [^3^H]Dopamine Release from Rat Striatum

After a 60-min initial washout period, the striatal tissue content of [^3^H]dopamine reached a value of 332.73 ± 82.28 kBq/g and this content of radioactivity declined to 205.65 ± 71.56 kBq/g during the following 75 min superfusion period. These data indicate that approximately 38% of tissue content of [^3^H]dopamine is released during superfusion (Figure 2A).

The resting [^3^H]dopamine release approached a rate of 1.92 ± 0.21 percent of total tissue content of radioactivity in 3 min determined after the 60-min initial washout period. When the superfused striatal slices were stimulated electrically, the release of [^3^H]dopamine increased from 1.66 ± 0.11 to 3.39 ± 0.43 percent of tissue [^3^H]dopamine content released in 3 min (*n* = 6, *p* < 0.01). These values correspond to a [^3^H]dopamine release of 5.15 ± 1.26 and 10.45 ± 2.96 kBq/g/3 min measured in resting condition and in response to electrical stimulation (Figure 2A). [^3^H]Dopamine release induced by electrical stimulation from striatal slices is an external Ca^2+^-dependent process originates from vesicles, and the release is a result of exocytosis [38].

[^3^H]Dopamine release can also be evoked by chemical stimulation. Chemically induced [^3^H]dopamine release originates from the cytoplasmic dopamine stores, it is external Ca^2+^-independent and the reverse dopamine transporter operation is involved in the release mechanism. An example of how drug action on non-vesicular [^3^H]dopamine release was determined is shown in Figure 2B.

### 2.2. Effect of Trace Amines, Releaser, and Enhancer Compounds on Non-Vesicular Release of [^3^H]Dopamine from Rat Striatum

Effects of a series of phenylalkylamines, which differ in the length of the alkyl chain substitutions on the nitrogen and the α carbon atoms, are shown in Figure 3A. The TAAR1 receptor endogenous ligand phenylethylamine and the exogenous ligand (±)methamphetamine, in which methyl substitutions occur instead of hydrogens on the nitrogen and the α carbon atoms, increased resting [^3^H]dopamine release. Gradual increase in length of these alkyl chains from methyl to ethyl or propyl groups resulted in the loss of releasing ability of the compounds (for chemical structures, see Figure 1). These experiments indicate a basic difference between releaser and enhancer compounds (i.e., (±) methamphetamine and (−)PPAP): the enhancer compounds did not exhibit an appreciable effect on resting [^3^H]dopamine release.

Figure 3B demonstrates that the enhancer compounds (−)PPAP, (−)IPAP, and (−)BPAP, which all contain propyl group substitutions on the nitrogen and α carbon atoms but with various rings attached (phenyl, indole, and benzofuran) to the side chain, do not possess releasing ability (for chemical structures see Figure 1. These experiments demonstrate that the alkyl substitutions on the side chain but not the ring attached determines the releaser or enhancer characteristics of the phenylethylamine derivatives used here.

### 2.3. Effects of the Catecholamine Releaser (±)Methamphetamine on Vesicular and Non-Vesicular [^3^H]Dopamine Release in Rat Striatum

In agreement with previously reported data, we found here that (±)methamphetamine increased resting [^3^H]dopamine release in the low μmolar range (Figure 3A and Figure 4A). Figure 4A demonstrates that the increase of resting [^3^H]dopamine release by (±)methamphetamine was a concentration-dependent process.

(±)Methamphetamine also increased the electrical stimulation-induced [^3^H]dopamine release from the striatum (Figure 4B). This finding indicates that the releaser (±)methamphetamine affects both the non-vesicular and vesicular dopamine release. The two release processes appear in the same concentration range (1–10 µmol/L), although they are mediated by different mechanisms.

### 2.4. Effects of the Catecholamine Activity Enhancer (−)BPAP on Vesicular and Non-Vesicular [^3^H]Dopamine Release in Rat Striatum

The enhancer effect of the CAE compound (−)BPAP (10^−12^ mol/L) on [^3^H]dopamine release determined in striatal slices is shown in Figure 5A,B. (−)BPAP increased the electrical stimulation-induced [^3^H]dopamine release compared to that measured in control conditions, this effect was not associated with alteration of the resting [^3^H]dopamine release.

In the next series of experiments, we added (−)BPAP in increasing concentrations to striatal slices. As shown in Figure 6A, (−)BPAP added in a concentration range of 10^−15^ to 10^−5^ mol/L elicited a biphasic effect: it increased electrical stimulation-induced release of [^3^H]dopamine in concentrations of 10^−12^ and 10^−11^ mol/L and in concentrations of 10^−6^ and 10^−5^ mol/L, respectively. The former effect was considered as a specific enhancer effect.

While (−)BPAP in concentrations of 10^−15^ to 10^−5^ mol/L increased electrical stimulation-induced [^3^H]dopamine release, it was without effect on resting release (Figure 6B). This finding suggests that (−)BPAP primarily alters dopamine release originating from vesicular stores and exerts only a marginal effect on non-vesicular dopamine release. The basic difference between the effects of the releaser and the enhancer compounds on dopaminergic neurochemical transmission is indicated by the findings that (±)methamphetamine evoked dopamine release from both cytoplasmic and vesicular stores whereas (−)BPAP affected only exocytotic release.

### 2.5. Effect of the Dopamine Transporter Inhibitor Nomifensine on [^3^H]Dopamine Release in Rat Striatum

The dopamine transporter (DAT) inhibitor nomifensine [39] increased the electrical stimulation-induced release of [^3^H]dopamine from striatal slices and this effect was found to be concentration-dependent in a range of 1 to 10 µmol/L (Figure 7A,B). These findings prove that electrical stimulation results in an elevation in extracellular concentrations of dopamine following inhibition of dopamine reuptake [40].

An increase in electrical stimulation-induced [^3^H]dopamine release was associated with unaffected resting [^3^H]dopamine release, indicating that nomifensine does not force DAT operation into the reverse mode (Figure 7B). Nomifensine is not carried into the cytoplasm of the nerve endings following its binding to DAT and a lack of its releasing activity was shown [41,42].

#### 2.5.1. Effect of Nomifensine on (±)Methamphetamine-Induced [^3^H]Dopamine Release

Using striatal slices in our experiments, we found that nomifensine (1–10 µmol/L) reversed the dopamine-releasing effect of (±)methamphetamine and this inhibitory effect proved to be concentration-dependent (Figure 8A,B). This finding supports the view that (±)methamphetamine as a transporter substrate inhibitor is taken up by DAT into the dopaminergic axon terminals, which is an essential initial step to evoke increased dopaminergic activity.

#### 2.5.2. Effect of Nomifensine on (−)BPAP-Induced [^3^H]Dopamine Release

Interaction between nomifensine and (−)BPAP on [^3^H]dopamine release was investigated in a condition when (−)BPAP (10^−11^ mol/L) was added in the presence of 1 µmol/L nomifensine in rat striatal slices (Figure 9). It was found that in the presence of nomifensine, (−)BPAP did not increase electrical stimulation-induced [^3^H]dopamine release. This finding suggests that (−)BPAP enters dopaminergic axon terminals by interaction with DAT, and the carrier operates in a nomifensine-sensitive uptake mode during the transport.

### 2.6. Effect of the TAAR1 Antagonist EPPTB on [^3^H]Dopamine Release in Rat Striatum

We tested the effect of EPPTB, the first described selective inhibitor of TAAR1 [8] per se on [^3^H]dopamine release and also its effects on the releaser (±)methamphetamine and the enhancer (−)BPAP-mediated [^3^H]dopamine release. EPPTB, added in a concentration range of 0.01 to 1.0 µmol/L, did not influence resting or electrical stimulation-induced [^3^H]dopamine release in rat striatum (Figure 10A,B). This suggests that TAAR1 does not exhibit agonist-independent constitutive activity in dopaminergic neurotransmission of the striatum.

#### 2.6.1. Effect of EPPTB on (±)Methamphetamine-Induced [^3^H]Dopamine Release

In an interaction study, EPPTB was added in a concentration of 1 µmol/L to striatal slices and reversed the releaser effect of (±)methamphethamine (10 µmol/L) on resting [^3^H]dopamine release (Figure 11A). Whereas EPPTB (0.1 and 1 µmol/L) effectively decreased the stimulatory effect of (±)methamphethamine (6.75 µmol/L) on resting [^3^H]dopamine release, it did not influence the stimulatory effect of (±)methamphetamine on electrical stimulation-induced [^3^H]dopamine release (Figure 11B). This finding indicates a different involvement of the TAAR1 receptor in the non-vesicular and vesicular release-inducing effects of (±)methamphetamine.

#### 2.6.2. Effect of EPPTB on (−)BPAP-Induced [^3^H]Dopamine Release

Since the TAAR1 antagonist EPPTB failed to modify the effect of (±)methamphetamine on electrical stimulation-induced [^3^H]dopamine release, an interaction study was made to determine whether EPPTB influences the enhancer effect of (−)BPAP on striatal [^3^H]dopamine release. EPPTB added in a concentration of 0.01 or 0.1 µmol/L suspended the stimulatory effect of (−)BPAP on electrical stimulation-induced [^3^H]dopamine release evoked by either 10^−12^ or 10^−11^ mol/L concentration in striatal slices (Figure 12A,B). This finding indicates that the enhancer compound (−)BPAP and the TAAR1 antagonist EPPTB interact, and (−)BPAP may bind to and stimulate TAAR1 in dopaminergic axon terminals and consequently the release.

### 2.7. The Role of the Protein Kinase-Mediated Phosphorylation in the Dopamine Releaser and Enhancer Drug Actions

To study the involvement of protein kinase C (PKC)-mediated phosphorylation in [^3^H]dopamine release, the PKC inhibitor Ro31-8220 and activator phorbol-12-myristate-13-acetate (PMA) were used [43]. We have found that the inhibitor and activator of PKC exerted opposite effects on the electrical stimulation-induced release of [^3^H]dopamine from striatal slices. Moreover, it was shown that the effects of the releaser and enhancer compounds on [^3^H]dopamine release were PKC phosphorylation-dependent processes.

#### 2.7.1. Effect of the Protein Kinase C Inhibitor Ro31-8220 on [^3^H]Dopamine Release in Rat Striatum: An Inhibition

The PKC inhibitor Ro31-8220 (0.1–10 µmol/L) added to striatal slices concentration-dependently decreased the electrical stimulation-induced release of [^3^H]dopamine (Figure 13A,B).

#### 2.7.2. Effect of Ro31-8220 on (±)Methamphetamine-Induced [^3^H]Dopamine Release

A series of PKC inhibitors (chelerythrine, calphostin C, and Ro31-8220) were reported to inhibit carrier-mediated amphetamine-induced dopamine release [44] and our experiments are in accordance with these previous findings. Thus, Ro31-8220 (1 µmol/L) suspended the stimulatory effect of 10 µmol/L (±)methamphetamine on resting [^3^H]dopamine release from rat striatum (Figure 14). This finding confirms that a PKC-mediated phosphorylation is an essential step in the mechanism of (±)methamphetamine to evoke non-vesicular dopamine release from dopaminergic axon terminals.

#### 2.7.3. Effect of Ro31-8220 on (−)BPAP-Stimulated [^3^H]Dopamine Release

Whether PKC-mediated phosphorylation is also involved in the enhancer effect, interactions of (−)BPAP with the PKC inhibitor Ro31-8220 and the PKC activator PMA were determined in striatal [^3^H]dopamine release. Ro31-8220 (1 µmol/L) suspended the stimulatory effect of (−)BPAP (10^−12^ mol/L) measured on electrical stimulation-induced [^3^H]dopamine release from rat striatal slices (Figure 15). This finding indicates that a PKC-mediated phosphorylation is a critical step in the mechanism of (−)BPAP to evoke vesicular dopamine release from dopaminergic axon terminals.

#### 2.7.4. Effect of the Protein Kinase C Activator Phorbol 12-Myristate 13-Acetate on [^3^H]Dopamine Release in Rat Striatum: A Stimulation

The PKC activator phorbol 12-myristate 13-acetate (PMA, 0.01–1 µmol/L) added to striatal slices concentration-dependently increased the electrical stimulation-induced release of [^3^H]dopamine (Figure 16). The estimated concentration of PMA that increased the electrical stimulation-induced [^3^H]dopamine release by 50% above control release was 0.8 µmol/L.

#### 2.7.5. Effect of Phorbol 12-Myristate 13-Acetate on (−)BPAP-Stimulated [^3^H]Dopamine Release

We also tested the PKC activator PMA to influence the enhancer effect of (−)BPAP on the electrical stimulation-induced [^3^H]dopamine release in striatal slices. It was found that PMA used in a concentration specific for PKC activation (1 μmol/L) potentiated the enhancer effect of (−)BPAP on electrical stimulation-evoked [^3^H]dopamine release when added in a 10^−12^ mol/L concentration (Figure 17). These findings suggest that PKC involved in both non-vesicular and vesicular [^3^H]dopamine release.

### 2.8. Effect of N-Ethylmaleimide-Sensitive Factor Inhibitor N-Ethylmaleimide on [^3^H]Dopamine Release in Rat Striatum

N-Ethylmaleimide (NEM), an inhibitor of the ATPase N-ethylmaleimide-sensitive factor (NSF), was shown to increase the size of the readily releasable pool and the subsequent dopamine release [43]. We found here that NEM, added to striatal slices by itself, did not evoke an increase in resting [^3^H]dopamine release but increased the electrical stimulation-induced [^3^H]dopamine release (Figure 18A,B). This effect of NEM was concentration dependent in a range of 10 to 100 µmol/L. The estimated concentration of NEM that increased [^3^H]dopamine release by 50% above control release was 12.5 µmol/L.

#### Effect of N-Ethylmaleimide on (−)BPAP-Stimulated [^3^H]Dopamine Release

When NEM was added in a concentration of 30 µmol/L in combination with (−)BPAP (10^−12^ mol/L), a clear-cut potentiation was observed in the electrical stimulation-induced [^3^H]dopamine release (Figure 19). This potentiation may be due to an activation by (−)BPAP of TAAR1 signaling and the increased PKC-mediated phosphorylation triggers upregulation of the readily releasable pool of dopamine [45].

### 2.9. The Role of the Vesicular Monoamine Transporter 2 in [^3^H]Dopamine Release in Rat Striatum: Effect of Tetrabenazine

Tetrabenazine (TBZ), a non-substrate-type, non-competitive inhibitor of vesicular dopamine transporter 2 (VMAT2), inhibits dopamine accumulation in the vesicles leading to depletion of vesicular dopamine content and decrease in exocytotic release [46]. In accordance with these findings, Figure 20 shows electrical stimulation-induced [^3^H]dopamine release from striatal slices obtained from saline, (−)BPAP), TBZ, and TBZ and (−)BPAP pretreated rats. TBZ pretreatment of rats reduced [^3^H]dopamine release and this reduction in release was reversed by concomitant injection of TBZ and (−)BPAP. Injection of (−)BPAP was without effect on [^3^H]dopamine release in these experimental conditions.

#### Effect of Phorbol 12-Myristate 13-Acetate on [^3^H]Dopamine Release in Tetrabenazine-Pretreated Rat Striatum

In the next series of experiments, we found that the PKC activator PMA suspends the TBZ-induced inhibition of [^3^H]dopamine release in rat striatum (Figure 21). This finding indicates that PKC-mediated phosphorylation may be a key step in VMAT2-mediated release from dopamine-containing vesicles. In fact, PKC-dependent phosphorylation of VMAT2 is necessary for the maintenance of monoamine uptake into the vesicles [47].

### 2.10. The Role of TAAR1 in D2 Receptor-Mediated Presynaptic Negative Feedback Inhibition

Dopamine released from axon terminals activates the presynaptic D2 receptors and the negative feedback inhibition results in decreased further release of dopamine [48]. We have used sulpiride, a D2 receptor antagonist, to suspend D2 dopamine receptor-mediated autoinhibition of [^3^H]dopamine release. Sulpiride increased the electrical stimulation-induced [^3^H]dopamine release and this increase indicated the activation rate of dopamine autoreceptor. Blocked of TAAR1 by the receptor antagonist EPPTB led to a decrease in sulpiride-stimulated [^3^H]dopamine release, indicating that D2 receptor-mediated autoinhibition requires operation of TAAR1 signaling (Figure 22).

## 3. Discussion

### 3.1. Effects of the Enhancer Compound (−)BPAP on [^3^H]Dopamine Release from Striatal Slices of the Rat

[^3^H]Dopamine release from striatal slices preloaded with radioactive substance occurs at a constant rate in resting conditions. Spontaneously leak and fusion of docked dopamine vesicles result in resting (spontaneous) neurotransmitter release in the absence of action potential [49,50]. Resting [^3^H]dopamine release can be stimulated by two mechanisms: reverse-mode operation of neurotransmitter transporters leads to non-vesicular release, and membrane depolarization-evoked exocytosis results in vesicular neurotransmitter release.

Our experiments confirmed previous findings showing that the enhancer compound (−)BPAP used in a wild concentration range increased the action potential-mediated [^3^H]dopamine release without altering resting release [25,51]. The stimulatory effect of (−)BPAP on the electrical stimulation-evoked [^3^H]dopamine release exhibited a biphasic shape. Although the biphasic effect of (−)BPAP was found in measuring radiolabeled dopamine release, this effect occurred parallel with the release of endogenous dopamine [26,28]. The first peak in dopamine release observed in response to (−)BPAP was detected in the low picomolar concentrations and was called the enhancer effect [25]. The concentrations of (−)BPAP that evoked a second peak in dopamine release, fell into the µmolar range and previous reports concluded that this effect of (−)BPAP is due to an inhibition of type A MAO [51].

The biphasic concentration-response effect of (−)BPAP to release [^3^H]dopamine following electrical stimulation suggests an interaction of (−)BPAP with two different dopamine release sites to which the enhancer compound may bind with high (picomolar) and low (µmolar) affinities. (−)BPAP acting on the high-affinity binding site may evoke dopamine release in a TBZ-sensitive manner [46]. Alternatively, the two dopamine release sites are linked to two distinct vesicular stores, the readily releasable dopamine store (small vesicles) and the long-term store (large vesicles) [52]. It is also possible that the two dopamine release sites may be linked to two pools of dopamine within the vesicles, a free pool and a pool associated with ATP complexes [53].

#### 3.1.1. Evidence That TAAR1 Is Involved in the Mechanism of Action of the Enhancer Compound (−)BPAP

The finding that (−)BPAP influenced vesicular dopamine release in the picomolar concentrations, raised the possibility of an effect on TAAR1 as this receptor is notorious in its activation by low concentrations of various endogenous and exogenous compounds [1,19,37]. Whether the enhancer compound (−)BPAP influences TAAR1 activity, a series of interaction studies were carried out by the addition of (−)BPAP and the TAAR1 inhibitor EPPTB in combinations [8,54]. It was found that EPPTB, used in specific concentrations, suspended the stimulatory effect of the low concentrations of (−)BPAP on electrical stimulation-evoked [^3^H]dopamine release. This finding raises the possibility that prior activation of TAAR1 is necessary for triggering the enhancer-mediated stimulation of vesicular [^3^H]dopamine release.

#### 3.1.2. Evidence That DAT Is Involved in Entry of the Enhancer Compound (−)BPAP into the Presynaptic Axon Terminals

Since TAAR1 is localized in the intracellular space, drugs are taken up into dopaminergic axon terminals before they interact with the receptor [21,55]. This transport of TAAR1 ligands is mediated by the plasma membrane transporter DAT. As the natural substrate of DAT is dopamine, other compounds that are carried in by this transporter, act as substrate inhibitors of DAT operation [56]. The effects of the enhancer compounds on DAT activity were investigated previously [57]. (−)BPAP was reported to inhibit dopamine uptake in HEK293 cells expressing hDAT in this study. Enhancer compounds, competing with dopamine for uptake at the transport site of DAT, are transported instead of the endogenous ligand dopamine; this effect appears as a weak dopamine uptake inhibition. Although (−)BPAP may be a substrate-type inhibitor of DAT, its binding to the transporter protein did not result in an increase of resting [^3^H]dopamine release when tested in a broad concentration range. This finding indicates that DAT operation, which bidirectionally induces uptake and release of dopamine, may be uncoupled in the presence of the enhancer drugs [58].

In contrast, the competitive DAT inhibitor nomifensine blocks transporter operation in both normal- and reverse-mode directions [39]. Nomifensine is not carried into the cytoplasm of the nerve endings following its binding to DAT and lacks dopamine releasing activity [41,42]. However, nomifensine inhibited (−)BPAP to stimulate electrically induced [^3^H]dopamine release by blocking DAT operation. This finding indicates that the enhancer drugs need first to be transported into dopaminergic axon terminals by DAT and bind to TAAR1 before they trigger vesicular dopamine release.

#### 3.1.3. Vesicular Dopamine Release following TAAR1 Activation: The Role of Intracellular Phosphorylation

The signal transduction coupled to TAAR1 involves a series of intracellular phosphorylation processes mediated by PKA and PKC [11,55]. This series of phosphorylation targets a number of proteins (SNAP-25, Munch18 protein) involved in the process of vesicular dopamine release [59]. The PKC pool connected to vesicles exhibits sensitivity to reserpine and is involved in the regulation of exocytotic dopamine release [60]. In our experiments, inhibition or activation of PKC exerted opposite effects on the electrical stimulation-induced [^3^H]dopamine release: the PKC inhibitor Ro31-8220 decreased and the PKC activator PMA increased the vesicular dopamine release. This finding confirms that exocytosis requires PKC-mediated phosphorylation of proteins involved in vesicular neurotransmitter release [43,45]. In contrast to vesicular release, resting (spontaneous) dopamine release was not altered by PKC-mediated phosphorylation.

The enhancer compounds, which are putative agonist ligands of TAAR1, may increase the phosphorylation of proteins involved in exocytotic dopamine release. This was shown in interaction studies demonstrating the enhancer effect of (−)BPAP in the presence of PMA and Ro31-8220. The stimulatory effect of (−)BPAP on vesicular [^3^H]dopamine release was potentiated in the presence of PMA, whereas the addition of Ro31-8220 reduced the enhancer effect. Based on this finding, we concluded that (−)BPAP activating TAAR1 and its signaling system increases PKC-mediated phosphorylation of proteins participating in exocytosis, which then evokes enhanced dopamine release from vesicular stores.

#### 3.1.4. The Role of SNARE Core Complex in the Enhancer Regulation of Dopamine Release

PKC-mediated phosphorylation upregulates docked vesicles at the active zone by inducing conversion of vesicles from docking to priming and further to fusion states [45]. As a result of upregulation, the depolarization-induced neurotransmitter release increases from the readily releasable pool [43]. This process corresponds to the assembly of the SNARE core complexes by the interaction of a series of vesicular and nerve ending membrane proteins (synaptobrevin and syntaxin, SNAP-25) [61]. On the other hand, the SNARE core complexes are dissociated by a specific ATPase, NSF with its cofactor α-SNAP, suspending the further release of neurotransmitters [43]. NSF can be inhibited by NEM, which decreases the dissociation rate of the complexes, enhances the size of readily releasable pool, and maintains the depolarization-induced neurotransmitter release. According to this, we found that both the PKC activator PMA and the NSF inhibitor NEM increased the electrical stimulation-induced vesicular [^3^H]dopamine release, which may correspond with the size of the readily releasable pool.

The assembly and disassembly of the SNARE core complexes may have a role in enhancer modulation of neurochemical transmission. This was indicated by the fact that the stimulatory effect of (−)BPAP on electrical stimulation-induced [^3^H]dopamine release was potentiated by the addition of PMA and NEM, respectively. We have hypothesized that the CAE compound (−)BPAP activates TAAR1 signaling and the increased PKC-mediated phosphorylation maintains SNARE core complexes assembled, which then turns on the enhancer effect in vesicular dopamine release.

#### 3.1.5. The Role of Vesicular Monoamine Transporter 2 in the Enhancer Regulation of Dopamine Release in Rat Striatum

The role of VMAT2 in dopaminergic axon terminals is vesicular refill and regulation of dopamine efflux from the vesicles into the cytoplasm [62]. This bidirectional uptake- and release-mode operation of VMAT2 is primarily determined by PKC-mediated phosphorylation at the N terminal of the transporter protein [47,63]. We have thus, speculated that activation of TAAR1 signaling and increase of PKC phosphorylation by the enhancer drugs may also target VMAT2 establishing a possible link between TAAR1 and vesicular functions.

The importance of VMAT2 phosphorylation in vesicular dopamine release was also shown in our experiments. We found that the non-competitive VMAT2 inhibitor TBZ inhibited [^3^H]dopamine release from striatal slices and this inhibition was probably due to a decrease in vesicular dopamine accumulation and contents [46,64]. The addition of the PKC activator PMA, however, suspended the TBZ-induced inhibition of the electrical stimulation-induced [^3^H]dopamine release, further pointing to the importance of VMAT2 phosphorylation in vesicular operation.

We have found that the TBZ-evoked decrease of [^3^H]dopamine release was overacted by in vivo administration of (−)BPAP in doses corresponding with its CAE effects. Therefore, we speculated that (−)BPAP activates TAAR1 signaling and the increased PKC-mediated phosphorylation of VMAT2 regulates vesicular traffic of dopamine. Our preliminary findings indicate that the enhancer compound (−)BPAP increases vesicular storage capability and elicits more vesicular dopamine released as electrical stimulation is applied.

### 3.2. Effects of the Releaser Compound (±)Methamphetamine on [^3^H]Dopamine Release from Striatal Slices of the Rat

The first major difference between the enhancer and releaser compounds was found in influencing resting dopamine release: whereas the releasers evoked an increase of resting [^3^H]dopamine release, this effect was absent when the enhancers were added to striatal slices. The second difference between the two groups of compounds was that although both the enhancer and releaser drugs stimulated vesicular [^3^H]dopamine release, this effect of the enhancers was linked to TAAR1 activation, whereas TAAR1 was apparently not involved in (±)methamphetamine-induced vesicular dopamine release.

#### 3.2.1. Differential Effect of TAAR1 in (±)Methamphetamine-Induced Vesicular and Non-Vesicular Dopamine Release

A number of publications indicate that amphetamines are exogenous agonists of TAAR1 [2,3]. (±)Methamphetamine that we used in our study as a representative of the amphetamine class, evoked increase of resting and electrical stimulation-induced [^3^H]dopamine release from rat striatal slices. The involvement of TAAR1 activation in (±)methamphetamine-induced [^3^H]dopamine release was shown by the finding that EPPTB reduced the (±)methamphetamine-induced non-vesicular [^3^H]dopamine release. In contrast, the stimulatory effect of (±)methamphetamine on electrically induced [^3^H]dopamine release was not altered by the addition of EPPTB. We concluded, therefore, that other than TAAR1 signaling may be responsible for the (±)methamphetamine-evoked vesicular dopamine release. Accordingly, TAAR1 regulates dopamine efflux both from the (±)methamphetamine-sensitive cytoplasmic pool and the enhancer-sensitive vesicular pool of dopamine in presynaptic nerve endings.

#### 3.2.2. Evidence That DAT Is Involved in Entry of (±)Methamphetamine into the Presynaptic Axon Terminals

Amphetamines compete with dopamine for transport at the uptake sites of DAT and their transport is preferred instead of the endogenous ligand dopamine [65]. This substrate inhibitory effect of amphetamines on DAT is accompanied by non-vesicular dopamine release due to the reverse operation of the transporter. Besides induction of the release mode of DAT, uptake inhibition also contributes to the total release [66]. As was shown earlier, the releaser compounds are taken up into the dopaminergic axon terminals by DAT before they evoke dopamine release. Previous experiments indicated that the competitive inhibitors of DAT suspended the dopamine-releasing effect of amphetamines [67,68]. The nomifensine that we used to inhibit DAT operation blocked not only the enhancer but also the releaser drugs to induce [^3^H]dopamine release. These data indicate that the releaser and enhancer drugs need first to be transported into dopaminergic axon terminals by DAT before they trigger non-vesicular and vesicular dopamine release, respectively.

#### 3.2.3. The Role of Intracellular Phosphorylation in Non-Vesicular Dopamine Release following TAAR1 Activation

It has been shown that dopamine uptake and efflux, mediated by normal- and reverse-mode operation of DAT, are phosphorylation-dependent processes [55,69]. This phosphorylation may involve different kinases: TAAR1-regulated inhibition of dopamine uptake involves both PKA and PKC, whereas the TAAR1-mediated dopamine efflux involves only the PKC pathway [12]. To study the involvement of PKC-mediated phosphorylation in non-vesicular [^3^H]dopamine release evoked by the releaser compounds, we used the PKC inhibitor Ro31-8220 in drug combination studies. It was found that Ro31-820 decreased (±)methamphetamine-induced non-vesicular [^3^H]dopamine release from rat striatum. In contrast, neither Ro31-8220, nor PMA affected resting dopamine release per se, suggesting that resting dopamine release, which may be due to leakage of dopamine-containing vesicles, does not require phosphorylation to occur.

It has been shown that two different PKC pools are present in the axon terminals: an amphetamine-sensitive pool is located in the synaptic plasma membrane and a reserpine-sensitive PKC pool is connected to the vesicles. The former is linked to DAT operation, whereas the latter is connected to exocytotic dopamine release [60]. Thus, amphetamines, following their transport to the cytoplasm, activate TAAR1 connected to a PKC pool and the phosphorylated DAT shifting the carrier operation into the reverse mode. For DAT-mediated non-vesicular dopamine efflux, phosphorylation of the transporter is essential [70].

#### 3.2.4. The Methamphetamine-Induced Vesicular Dopamine Release: The Role of Vesicular Monoamine Transporter 2

In agreement with reported data, we also found that (±)methamphetamine increased not only the resting but the electrical stimulation-induced [^3^Hdopamine release also [71,72,73]. The two release processes, although they appeared in the same concentration range (1–10 µmol/L), are mediated by different mechanisms [65,74]. Since the stimulatory effect of (±)methamphetamine on electrically induced [^3^H]dopamine release was not altered by EPPTB, we concluded that other than TAAR1 signaling may be responsible for the (±)methamphetamine-induced vesicular dopamine release.

Amphetamines, which are taken up by DAT into the cytoplasm, further accumulate in the vesicles by VMAT2 operation. These compounds exhibit a lipophilic weak base character and accept protons within the vesicles, decreasing the retention of stored dopamine. Dopamine effluxed from the vesicle storage sites and redistributed between the vesicles, the cytoplasm, and the large and small vesicles in the nerve endings [52,53,75]. Since the small dopamine vesicles form the readily releasable pool, electrical stimulation may induce strengthened dopamine release in the presence of amphetamines [58,72].

### 3.3. The TAAR Signaling: A Central Regulator of Presynaptic Dopaminergic Neurotransmission Events

TAAR1 is an intracellularly localized receptor with a signal transduction pathway that involves Gs protein, adenylyl cyclase, cAMP production, and PKA- and PKC-mediated intracellular phosphorylation processes [12] (Figure 23). Activation of TAAR1 may induce phosphorylation of a series of proteins involved in dopamine release and storage. TAAR1 signaling pathway suggests an interaction between cAMP-dependent pathways and PKC-mediated phosphorylation, possibly involving novel PKCs such as PKCε [76,77].

Our experiments carried out with releaser and enhancer compounds led us to the conclusion that TAAR1 possesses a central role in the regulation of neurochemical transmission at presynaptic level. This regulation controls DAT operation, non-vesicular and vesicular dopamine release, assembly of SNARE core complexes, vesicular accumulation of dopamine by VMAT2, and D2 dopamine autoreceptor-mediated feedback inhibition. Using activator and inhibitor drugs of PKC, we found evidence that this phosphorylation is a key step in DAT-mediated non-vesicular dopamine release. PKC activation also enhances vesicular dopamine release in response to impulse propagation as well as its accumulation in storage vesicles.

Differences observed in dopamine-releasing effects of the releaser and enhancer drugs raise the possibility that these drugs bind to different binding sites on TAAR1. We hypothesized that the substituents of the α carbon and nitrogen atoms in the side chains of these compounds determine this binding process. Thus, compounds containing proton or methyl group on α carbon and nitrogen atoms (PEA and (±)methamphetamine) bind to a binding site on TAAR1, triggering non-vesicular dopamine release. On the other hand, compounds containing the more bulky ethyl or propyl alkyl groups in the same positions (DEA and PPAP) preferably bind to another binding site on TAAR1, inducing vesicular [^3^H]dopamine release. (−)BPAP, which was used as a model compound of the enhancers, apparently binds to the latter hypothetical binding site on TAAR1, regulating vesicular [^3^H]dopamine release. We have also speculated that the two binding sites activate different PKC-mediated phosphorylations linked to TAAR1 resulting in that one binding site evokes non-vesicular and the other binding site evokes vesicular dopamine release [60].

Experiments carried out in cell lines coexpressing TAAR1 and DAT indicated that drugs, which alter TAAR1 activity, are taken up by this carrier and reach the receptor by intracellularly directed transport [78]. There is a reciprocal regulation between TAAR1 and DAT operation: TAAR1 activation leads to inhibition of the uptake mode and facilitation of the release mode of the transporter [55]. PKC-mediated phosphorylation of DAT is responsible for the latter event [12]. The non-vesicular dopamine release evoked by releaser compounds (amphetamines) can be suspended by PKC and TAAR1 inhibitors, respectively. Our experiments indicate, however, that TAAR1 is not involved in (±)methamphetamine-evoked vesicular dopamine release.

Experiments using the enhancer compound (−)BPAP served as direct evidence for the role of TAAR1 in vesicular (exocytotic) dopamine release, an effect that was previously questioned [6]. This was indicated by the fact that (−)BPAP stimulated vesicular dopamine is released in a TAAR1-dependent manner. Moreover, the mechanism of TAAR1 to regulate vesicular dopamine release was found to be PKC-dependent and induced phosphorylation of proteins of the SNARE core complex and VMAT2.

A functional link has been reported between TAAR1 and D2 dopamine receptor-mediated feedback inhibition in nerve endings and somatodendritic areas of dopaminergic neurons [6,13,79]. Agonist ligands of TAAR1 reduced neuronal firing rate by activating a potassium-mediated outward current, whereas the TAAR1 antagonist EPPTB increased dopaminergic neuronal firing in the ventral tegmental area [8]. We propose that the enhancer compound (−)BPAP evokes TAAR1-mediated vesicular dopamine release and it consequently activates feedback inhibition protecting dopaminergic neurons from an overstimulated state [74].

## 4. Materials and Methods

### 4.1. Animals and Drug Treatments

All experimental procedures were approved by the local Ethical Committees and were in accordance with the NIH Guide for the Care and Use of Laboratory Animals, 8th Edition, 2011. Male Wistar rats weighing 180–220 g were used for experiments. The animals were housed five to a cage in a temperature and humidity-controlled animal facility on a 12 h light/dark cycle (6.00 a.m. on, 6.00 p.m. off) with food and water available ad libitum. All efforts were made to minimize the harm to animals. All housing and experiments were performed in accordance with the European Communities Council Directives (2010/63/EU), the Hungarian Act for the Protection of Animals in Research (XXVIII.tv. 32.§). Animal care and handling protocols were approved by the regional animal health authority in Hungary (Pest County Government Office, resolution number: PE/EA/285-5/2020 (date: 19 March 2020).

When in vivo drug treatment was used, rats were treated with saline or drugs (tetrabenazine 1 mg/kg sc, (−)BPAP 0.0001 mg/kg sc) 60 min before the experiments.

### 4.2. Preparation of Rat Brain Slices

Rats were decapitated by a guillotine and the brains were removed from the skull. The striata were prepared according to Glowinsky and Iversen [80] and slices from the striatum were prepared by a McIlwan tissue chopper (Ted Pella Inc., Redding, CA, USA). The brain slices were collected and immersed in oxygenated (95% O_2_/5% CO_2_) Krebs-bicarbonate buffer (composition in mmol/L: NaCl 118, KCl 4.7, CaCl_2_ 1.25, NaH_2_PO_4_ 1.2, MgCl_2_ 1.2, NaHCO_3_ 25, glucose 11.5, ascorbic acid 0.3, and Na_2_EDTA 0.03) at room temperature.

### 4.3. Release of [^3^H]Dopamine from Rat Brain Slices

Rat brain slices were loaded with [^3^H]dopamine (10 µCi) for 30 min in 1.5 mL aerated (95% O_2_/5% CO_2_, pH 7.4) and preheated (37 °C) Krebs-bicarbonate buffer [81]. After loading the tissues with [^3^H]dopamine, the brain slices were transferred into low volume (0.3 mL) superfusion chambers (Experimetria Kft, Budapest, Hungary) and superfused with aerated and preheated Krebs-bicarbonate buffer. The flow rate was kept at 1 mL/min by a Gilson multichannel peristaltic pump (type M312, Villiers-Le Bel, France). The superfusate was discarded for the first 60 min period of the experiments, then twenty-five 3-min fractions were collected by a Gilson multichannel fraction collector (type FC-203B, Middletown, WI, USA). When used, biphasic electrical field stimuli (40 V voltage, 10 Hz frequency, 2-msec impulse duration for 3 min in fractions 4 and 18) were delivered by a Grass S88 Electrostimulator (Quincy, MA, USA) to evoke [^3^H]dopamine release. Drugs were added to brain slices, as is indicated in the Figure legends and were maintained throughout the experiments.

### 4.4. Determination of [^3^H]Dopamine Efflux

At the end of superfusion, tissues were collected from the superfusion chambers, weighed, and solubilized in 0.4 mL Soluene-350. An aliquot (50 µL) was mixed with 5 mL of liquid scintillation reagent (Ultra Gold XR) and subjected to liquid scintillation spectrometry for the determination of tissue content of radioactivity. The tissue content of [^3^H]dopamine was expressed as kBq/g tissue.

To determine the radioactivity released from brain slices, a sample (1 mL) of the superfusate was mixed with 5 mL of liquid scintillation reagent and subjected to liquid scintillation spectrometry. The efflux of [^3^H]dopamine was expressed in kBq/g/3 min fraction or as a fractional rate, i.e., a percentage of the amount of radioactivity in the tissue at the time of the release was determined. To estimate the electrically induced [^3^H]dopamine overflow, the mean of the basal outflow determined before and after stimulation was subtracted from each sample and summed [81].

The effect of drugs on resting [^3^H]dopamine release was determined in the presence and absence of drugs and was expressed by the ratio of [^3^H]dopamine efflux in fraction 17 (presence of drug, B2) and fraction 3 (absence of drug, B1), i.e., B2/B1 ratio. The effects of drugs on electrically stimulated [^3^H]dopamine release were expressed by the ratio of [^3^H]dopamine efflux determined in response to the 2nd (presence of drug, S2) and 1st (absence of drug, S1) stimulations, i.e., S2/S1 ratio (Figure 2A). The Quattro Pro and the GraphPad Prism computer programs were used for data calculation.

### 4.5. Statistical Analyses

The Student paired *t*-test, the Student *t*-statistics for two-means, and the one-way ANOVA followed by the Dunnett’s test were used for statistical analysis of the data as appropriate. The mean ± S.E.M. was calculated and the number of independent determinations was indicated with n. A level of probability (p) less than 5% was considered significant.

### 4.6. Materials

[^3^H]Dopamine (dihydroxyphenylethylamine-3,4[^3^H], specific activity: 27.8 Ci/mmol), Soluene-350 tissue solubilizer, and Ultima Gold XR liquid scintillation reagent were obtained from PerkinElmer Life and Analytical Sciences, Boston, MA, USA). (−)BPAP (R-(−)-1-(benzofuran-2-yl)-2-propyl-aminopentane HCl) was obtained from Fujimoto Pharmaceutical Co., Osaka, Japan. Other enhancer drugs (−)PPAP (R-(−)-1-(phenyl-2-yl)-2-propylaminopentane HCl), (−)IPAP (R-(−)-1-(indol-3-yl)-2-propylaminopentane)) were also donated by the Fujimoto Pharmaceutical Co., EPPTB (Ro5212773, N-(3-ethoxyphenyl)-4-pyrrolidin-1-yl-3-trifluoromethylbenzamide), tetrabenazine, nomifensine maleate, and β-phenylethylamine HCl (PEA) were purchased from Sigma-Aldrich Chemical Co, Budapest, Hungary, 3-hydroxytyramine HCl was purchased from Merck, Germany. Ro31-8220 mesylate and phorbol 12-myristate 13-acetate (PMA) were obtained from Bio-Techne R and D Kft, Budapest, Hungary. N-Ethylmaleimide (NEM) was a product of Tokyo Chemical Industry Co., Ltd., Hungary. N,α-Diethylphenethylamine HCl (DEA) was purchased from Toronto Research Chemicals, North York, ON, Canada. (±)Amphetamine HCl and (±)methamphetamine HCl were received from Semmelweis University, Budapest, Hungary. All other chemicals were of analytical grade.

## 5. Conclusions

Activation of TAAR1 signaling increases dopamine release by either reversal of DAT operation or facilitation of exocytotic vesicular release. Enhanced dopamine release activates D2 dopamine receptor-mediated feedback inhibition in order to protect dopaminergic neurons from overload. We also concluded that the catecholaminergic activity enhancer drug (−)BPAP acts as an agonist on TAAR1, revealing some novel aspects in the regulation of presynaptic dopaminergic neurochemical transmission. Therapeutic aspects of the enhancer compounds, however, remain to be elucidated. Evidence has been obtained about the potential use of the CAE compounds as prophylactic agents in neurodegenerative disorders or in the treatment of the early phase of Parkinson’s and Alzheimer’s diseases. In addition, depression, mood disorders, various forms of anxieties, and the favorable influence of life expectancy or tumor genesis may represent further clinical applications of the enhancer compounds.

## Figures and Tables

**Figure 1 ijms-23-08543-f001:**
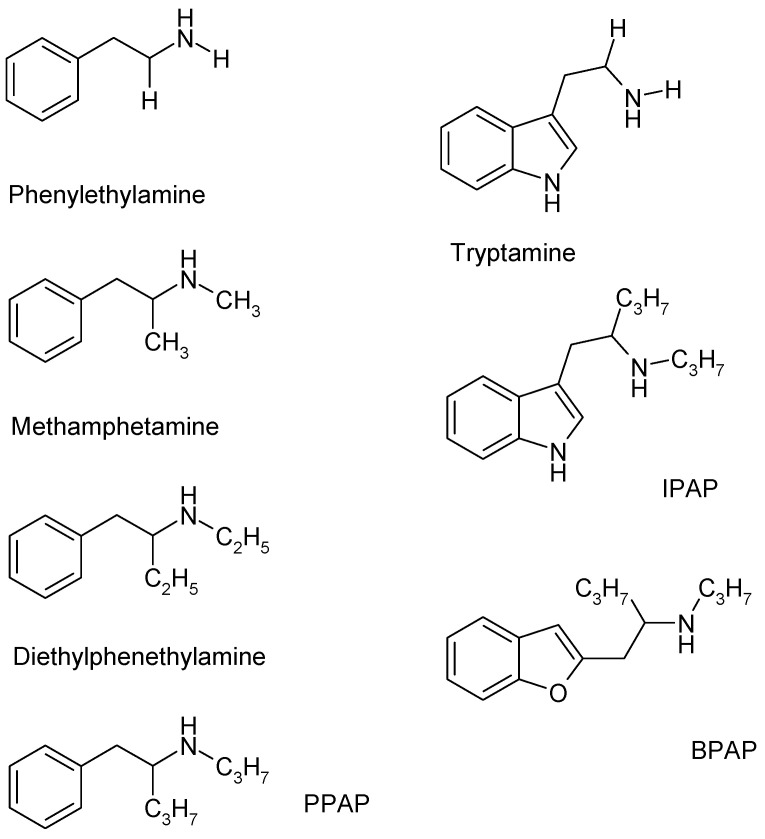
The chemical structure of trace amines (phenylethylamine, tryptamine), phenylethylamine analogues (methamphetamine, diethylphenethylamine), and enhancer substances (PPAP, IPAP, BPAP).

**Figure 2 ijms-23-08543-f002:**
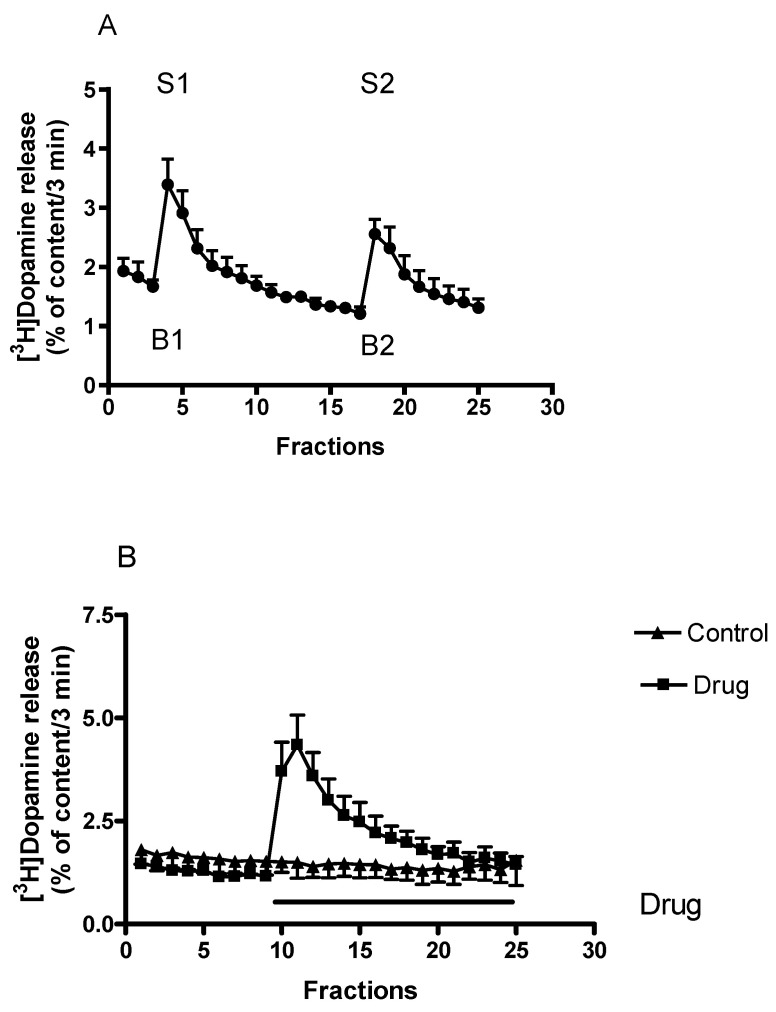
(**A**) The time-course of [^3^H]dopamine release measured from rat striatum. Filled circles indicate [^3^H]dopamine release at rest and in response to electrical stimulation. Striatal slices were prepared, loaded with [^3^H]dopamine and superfused with aerated Krebs-bicarbonate buffer. [^3^H]Dopamine release was induced by electrical stimulation (40 V, 2 Hz, 2 msec for 3 min) in fractions 4 (S1) and 18 (S2) and was expressed as a fractional rate, i.e., a percentage of the amount of [^3^H]dopamine in the tissue at the time of the release. The calculated ratio of the electrically stimulated fractional release S2 (2nd stimulation) over fractional release S1 (1st stimulation) (S2/S1) was 0.84 ± 0.04, representing a release of vesicular origin. The calculated ratio of resting fractional release B2 (fraction17) over fractional release B1 (fraction 3) (B2/B1) was 0.72 ± 0.05. When studied, drugs were added to the superfusion buffer between the 1st and 2nd electrical stimulations and maintained through the experiment, mean ± S.E.M., *n* = 6. (**B**) The time-course of non-vesicular [^3^H]dopamine release from rat striatum. Striatal slices were prepared, loaded with [^3^H]dopamine and superfused with aerated Krebs-bicarbonate buffer. The release of [^3^H]dopamine was expressed as a fractional rate expressed as percent of content released. [^3^H]Dopamine release was induced by the addition of drugs to the superfusion buffer and maintained through the experiment. In this experiment, (±)amphetamine as a drug was added in a concentration of 10 µmol/L from fraction 10 and the evoked non-vesicular [^3^H]dopamine release was 13.88 ± 3.00 percent of content released (calculated between fractions 10 and 25). The calculated [^3^H]dopamine release was 0.53 ± 0.12 percent of content released in control conditions. Student’s *t*-statistics for two-means, *p* < 0.01, mean ± S.E.M., *n* = 4-4.

**Figure 3 ijms-23-08543-f003:**
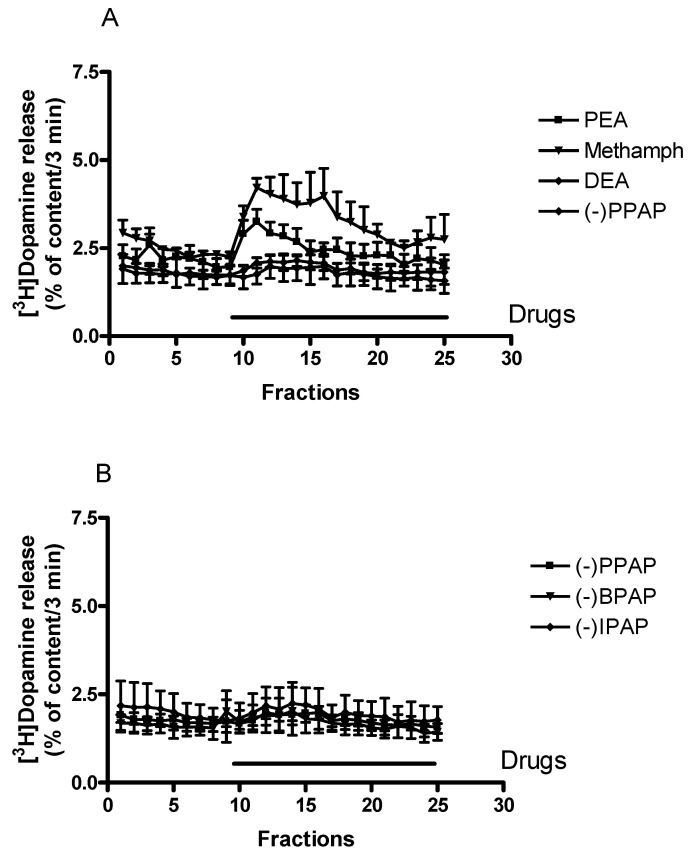
Effects of phenylethylamine derivatives on resting [^3^H]dopamine release from rat striatum. For the experimental procedure, see Figure 2B. (**A**) The trace amine phenylethylamine (PEA) and the releaser (±)methamphetamine (methamph) induced non-vesicular [^3^H]dopamine release, whereas N,α-diethylphenylethylamine (DEA) and the enhancer compound (−)PPAP were without effect on [^3^H]dopamine release. Drugs were added to rat striatal slices from fraction 10 in a concentration of 10 µmol/L, ((±)methamphetamine 6.75 µmol/L) and maintained through the experiment, mean ± S.E.M., *n* = 4. (**B**) The enhancer compounds (−)PPAP, (−)IPAP, and (−)BPAP were without effect on non-vesicular [^3^H]dopamine release. Drugs were added to rat striatal slices from fraction 10 in a concentration of 10 µmol/L and maintained through the experiment, mean ± S.E.M., *n* = 4. The control for Figure 3A,B is shown in Figure 2B.

**Figure 4 ijms-23-08543-f004:**
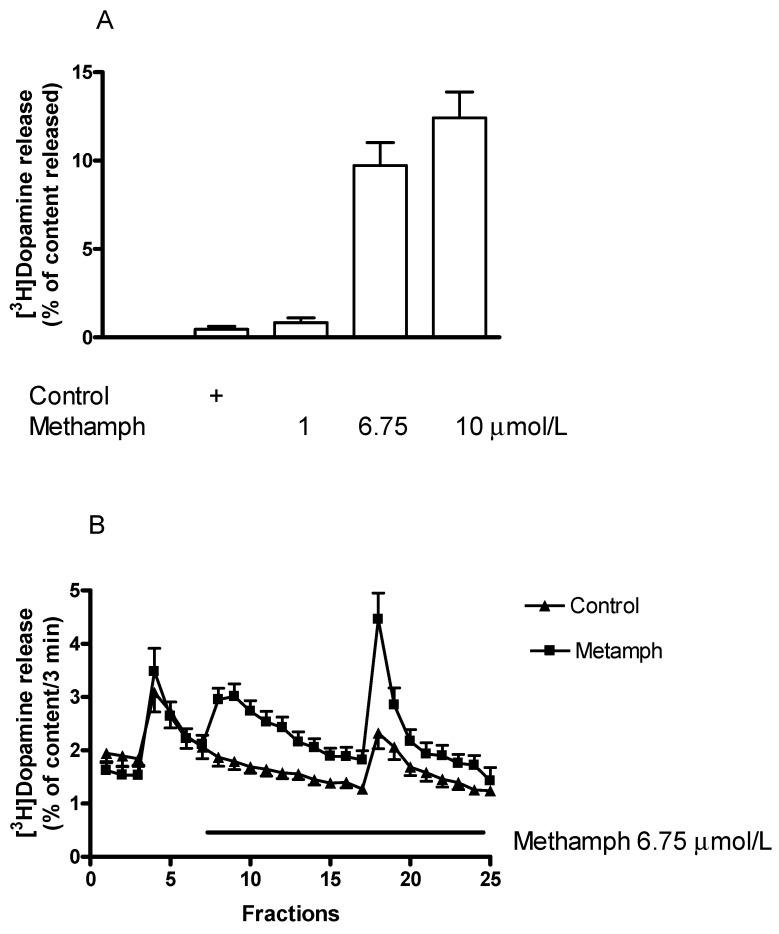
A (±)Methamphetamine (methamph) increased both non-vesicular and vesicular [^3^H]dopamine release from rat striatum. For the experimental procedure, see Figure 2A,B. (**A**) The releasing effect of (±)methamphetamine on resting [^3^H]dopamine efflux was concentration-dependent in a range of 1 to 10 µmol/L. The estimated concentration of (±)methamphetamine that increased resting [^3^H]dopamine release by 50% above control release was 3.2 µmol/L. One-way ANOVA followed by the Dunnett’s test, F(3,20) = 35.14, *p* < 0.001, (±)methamphetamine in 6.75 and 10 μmol/L concentrations significantly increased non-vesicular [^3^H]dopamine release, *p* < 0.01, mean ± S.E.M., *n* = 4-8. (**B**) (±)Methamphetamine was added to striatal slices from fraction 8 in a concentration of 6.75 µmol/L and maintained throughout the experiment. Resting [^3^H]dopamine release (defined as fractional release between fractions 8 and 17) was 0.37 ± 0.08 in control and 4.27 ± 0.46 percent of the content in the presence of (±)methamphetamine, *p* < 0.001. The electrical stimulation-induced [^3^H]dopamine release (S2/S1) was 0.89 ± 0.03 in control and 1.86 ± 0.32 in response to (±)methamphetamine, *p* < 0.01. Student *t*-statistics for two-means, mean ± S.E.M., *n* = 8-8.

**Figure 5 ijms-23-08543-f005:**
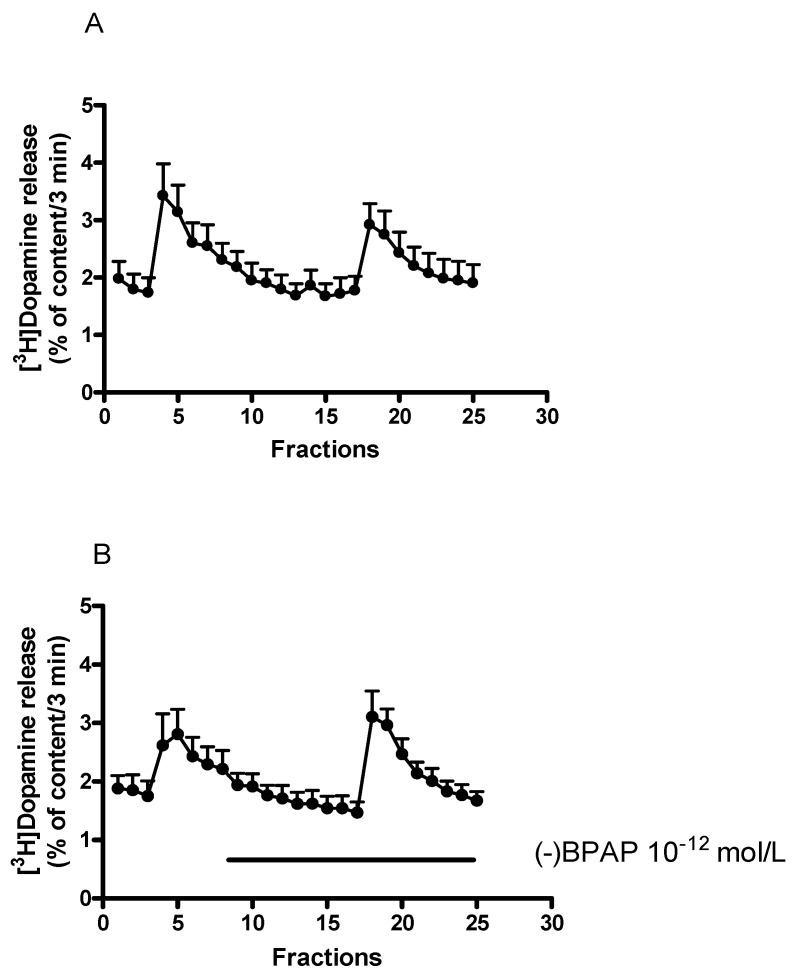
(−)BPAP increased electrical stimulation-induced [^3^H]dopamine release from rat striatum. For the experimental procedure, see Figure 2A. When studied, (−)BPAP was added in a concentration of 10^−12^ mol/L to the superfusion buffer from fraction 8 and maintained throughout the experiment. (**A**) The release of [^3^H]dopamine from striatal slices, control experiments. Filled circles indicate [^3^H]dopamine release at rest and in response to electrical stimulation. The resting [^3^H]dopamine release (B2/B1) and the electrical stimulation-induced [^3^H]dopamine release (S2/S1) were 0.92 ± 0.05 and 0.74 ± 0.04 in control conditions. (**B**) Effect of (−)BPAP (10^−12^ mol/L) on [^3^H]dopamine release from rat striatum. Filled circles indicate [^3^H]dopamine release in the presence and absence of (−)BPAP. (−)BPAP was added to striatal slices from fraction 8 and maintained throughout the experiment. The resting [^3^H]dopamine release (B2/B1) and the electrical stimulation-induced [^3^H]dopamine release (S2/S1) were 0.96 ± 0.08 and 1.24 ± 0.13 in the presence of (−)BPAP. These values indicate that (−)BPAP in 10^−12^ mol/L concentration failed to influence the resting (*p* = 0.55) but increased the electrically induced [^3^H]dopamine release (*p* < 0.01). Student *t*-statistics for two-means, mean ± S.E.M., *n* = 7-6.

**Figure 6 ijms-23-08543-f006:**
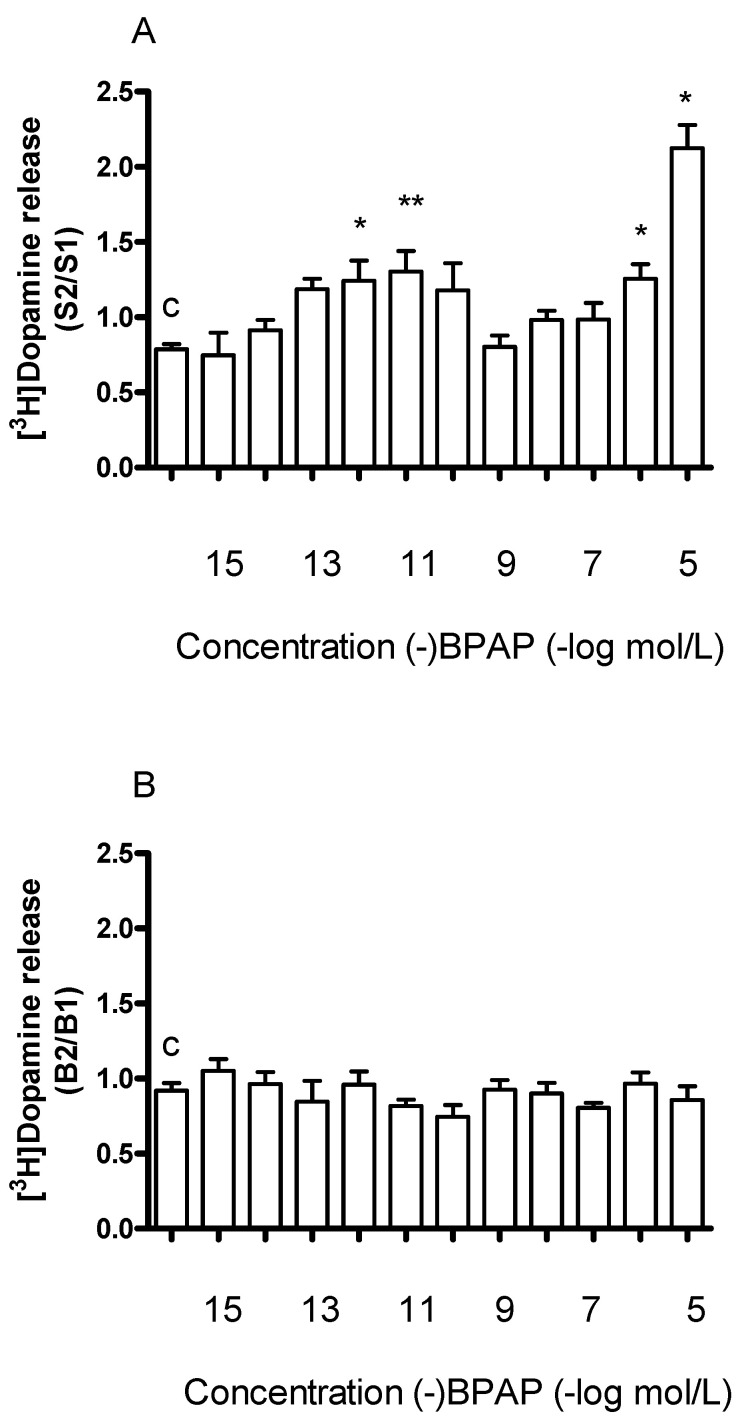
Concentration-dependent effect of (−)BPAP on resting and electrical stimulation-induced [^3^H]dopamine release from rat striatum. For the experimental procedure, see Figure 2A. (−)BPAP was added in a concentration range from 10^−15^ to 10^−5^ mol/L to the superfusion buffer from fraction 8 and maintained through the experiment. (**A**) The S2/S1 ratio indicates the effect of (−)BPAP on electrical stimulation-induced [^3^H]dopamine release determined in 1st (absence of drug, S1) and 2nd (presence of drug, S2), stimulations were carried out in fractions 4 and 18. The S2/S1 value was 0.79 ± 0.04 in control experiments (c). One-way ANOVA followed by the Dunnett’s test, F(11,48) = 10.16, *p* < 0.0001, * *p* < 0.05, ** *p* < 0.01, mean ± S.E.M., *n* = 4-8. (**B**) The B2/B1 ratio indicates the effect of (−)BPAP on resting fractional [^3^H]dopamine release determined in fractions 3 (absence of drug, B1) and 17 (presence of the drug, B2). The B2/B1 value was 0.91 ± 0.05 in control experiments (c). One-way ANOVA followed by the Dunnett’s test, F(11,48) = 1.171, *p* = 0.331, mean ± S.E.M., *n* = 4-8.

**Figure 7 ijms-23-08543-f007:**
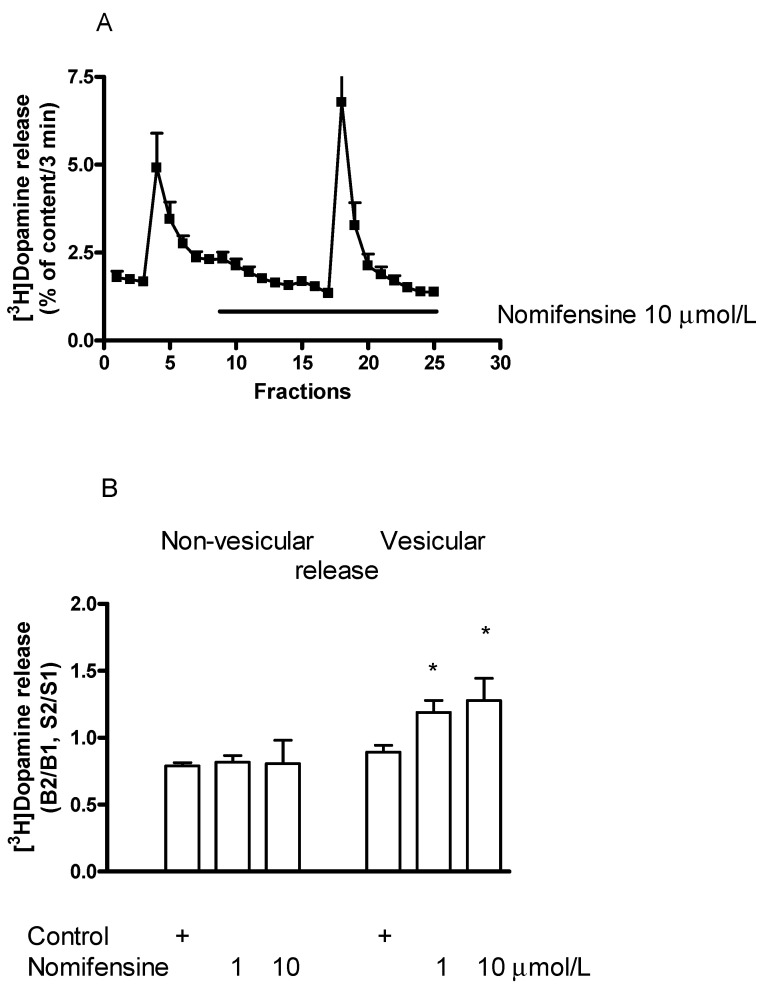
Effect of nomifensine on [^3^H]dopamine release from rat striatum. For the experimental procedure, see Figure 2A. (**A**) Filled circles indicate [^3^H]dopamine release in the presence and absence of nomifensine. Nomifensine was added to striatal slices from fraction 8 in a concentration of 10 µmol/L and maintained throughout the experiment. The B2/B1 ratio for resting release was 0.78 ± 0.02 in control and it was 0.80 ± 0.07 in the presence of nomifensine, not differing significantly. The S2/S1 ratio for electrical stimulation-induced release was 0.89 ± 0.05 in control and nomifensine increased this release to 1.27 ± 0.16 (*p* < 0.05). Student *t*-statistics for two-means, mean ± S.E.M., *n* = 8-4. (**B**) Nomifensine concentration dependently increased the electrically induced (vesicular) but not the resting (non-vesicular) [^3^H]dopamine release in striatal slices. One-way ANOVA followed by the Dunnett’s test, F(2,16) = 0.212, *p* = 0.812 for resting release and F(2,16) = 4.993, *p* = 0.02, * *p* < 0.05 for electrically stimulated release, mean ± S.E.M., *n* = 4-8.

**Figure 8 ijms-23-08543-f008:**
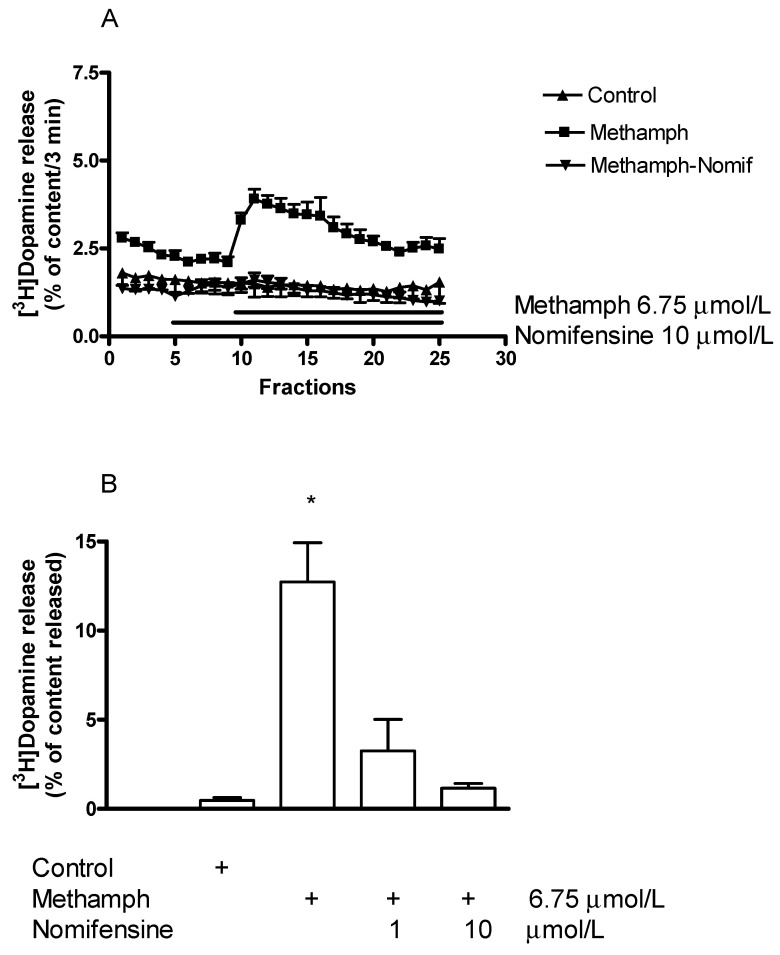
(**A**) Nomifensine (Nomif) reversed the (±)methamphetamine (methamph)-induced [^3^H]dopamine release from rat striatum. For the experimental procedure, see Figure 2B. (±)Methamphetamine was added to striatal slices from fraction 10 in a concentration of 6.75 µmol/L, nomifensine was added to the slices from fraction 5 in a concentration of 10 µmol/L and the drugs were maintained throughout the experiment. The (±)methamphetamine-induced [^3^H]dopamine release was 12.73 ± 2.20 and nomifensine decreased this release to 1.15 ± 0.26 percent of content (*p* < 0.01). Student *t*-statistics for two means, mean ± S.E.M., *n* = 4-4. (**B**) Nomifensine inhibited (±)methamphetamine-induced non-vesicular [^3^H]dopamine release in a concentration-dependent manner. (±)Methamphetamine was added to striatal slices from fraction 10 in a concentration of 6.75 µmol/L, nomifensine was added to the slices in a concentration of 1 or 10 µmol/L from fraction 1 and the drugs were maintained throughout the experiment. One-way ANOVA followed by the Dunnett’s test, F(3,12) = 15.58, *p* = 0.0002, * *p* < 0.05 mean ± S.E.M., *n* = 4.

**Figure 9 ijms-23-08543-f009:**
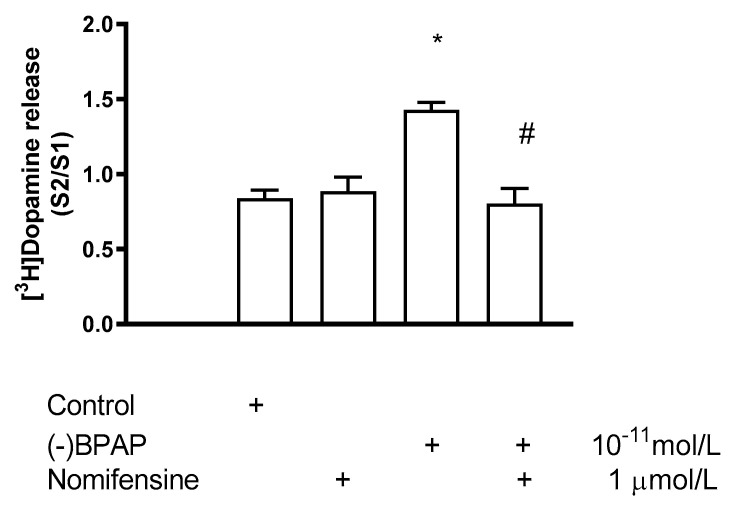
Effect of nomifensine on (−)BPAP-induced [^3^H]dopamine release in rat striatum. For the experimental procedure, see Figure 2A. (−)BPAP was added to striatal slices from fraction 8 in a concentration of either 10^−11^ mol/L in the presence and absence of nomifensine. When used, nomifensine was added to striatal slices from fraction 1 in a concentration of 1 µmol/L and drugs were maintained throughout the experiment. One-way ANOVA followed by the Dunnett’s test, F(3,12) = 4.784, *p* = 0.020, control vs. (−)BPAP effect * *p* < 0.05. Student *t*-statistics for two-means, (−)BPAP vs. (−)BPAP and nomifensine effect # *p* < 0.01, mean ± S.E.M., *n* = 4.

**Figure 10 ijms-23-08543-f010:**
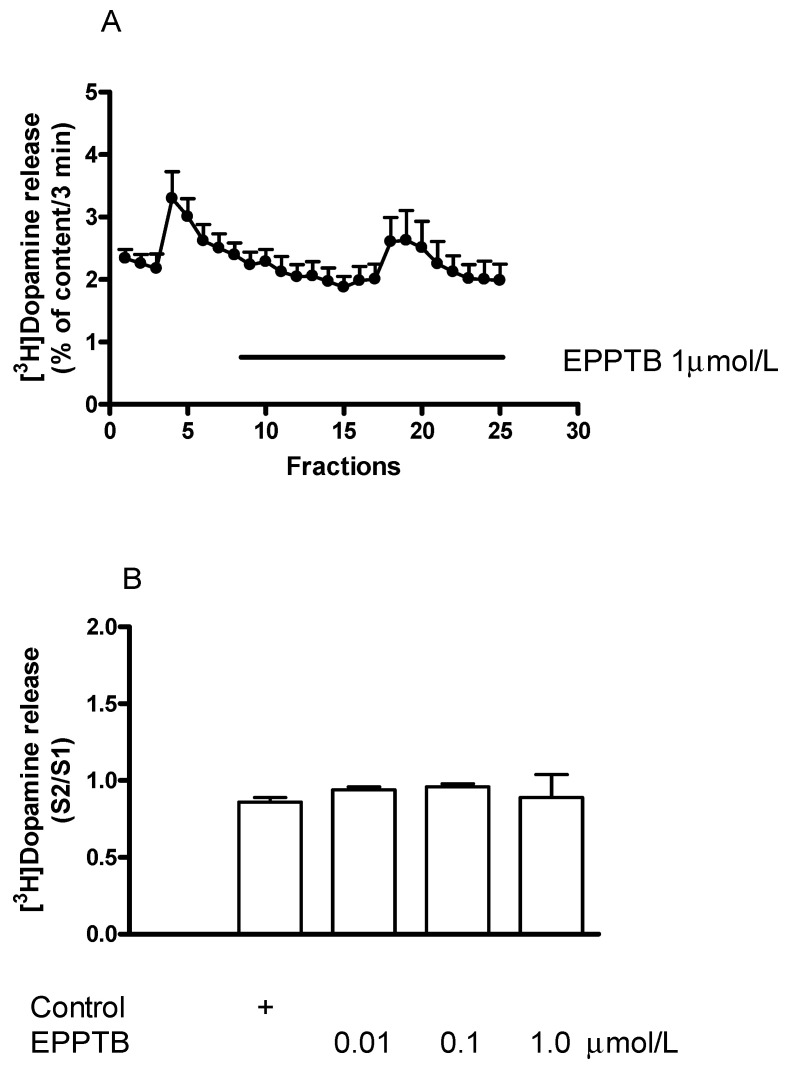
Effect of EPPTB on [^3^H]dopamine release from rat striatum. For the experimental procedure, see Figure 2A. (**A**) Filled circles indicate [^3^H]dopamine release in the presence and absence of EPPTB. EPPTB was added to striatal slices from fraction 8 in a concentration of 1 µmol/L and maintained through the experiment. The B2/B1 ratio for resting release was 0.85 ± 0.05 in control and it was 0.88 ± 0.07 in the presence of EPPTB, not differing significantly. The S2/S1 ratio for electrical stimulation-induced release was 0.94 ± 0.02 in control and it was 0.95 ± 0.03 in the presence of EPPTB. This difference was not significant: Student *t*-statistics for two-means, *p* > 0.05, mean ± S.E.M., *n* = 3-3. (**B**) Concentration-dependent effect of EPPTB on electrical stimulation-induced [^3^H]dopamine release in rat striatum. EPPTB was added to striatal slices from fraction 8 in concentrations varied from 0.01 to 1 µmol/L and maintained through the experiment. One-way ANOVA followed by the Dunnett’s test, F(3,13) = 1.757, *p* = 0.204, mean ± S.E.M., *n* = 3-8.

**Figure 11 ijms-23-08543-f011:**
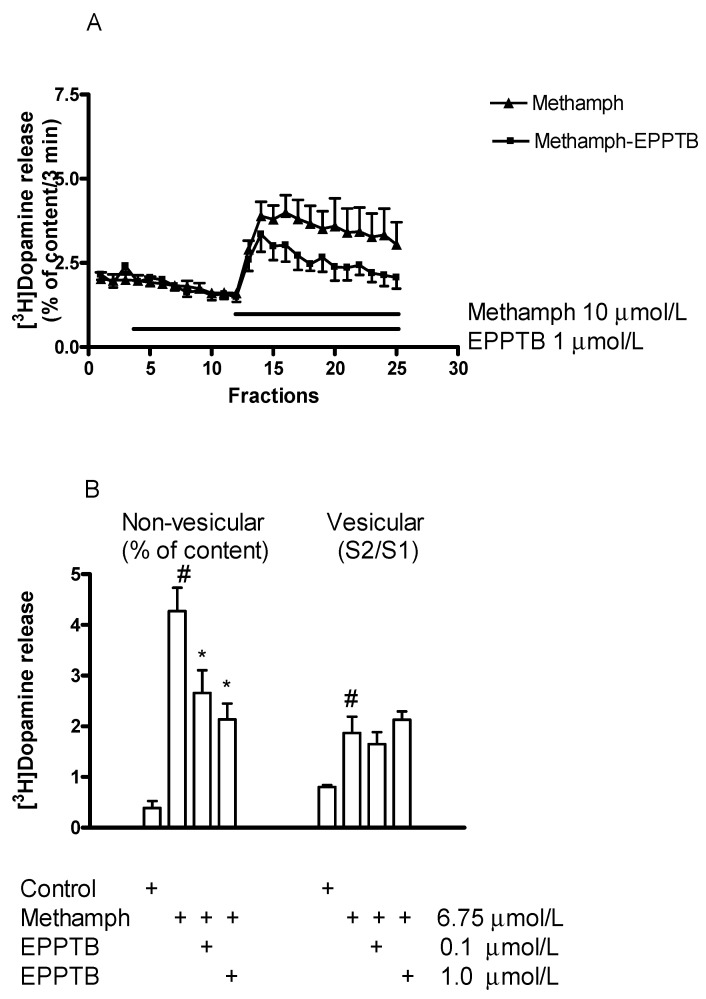
(**A**) EPPTB decreased the (±)methamphetamine (methamph)-induced non-vesicular [^3^H]dopamine release in rat striatum. For the experimental procedure, see Figure 2B. (±)Methamphetamine (10 µmol/L) was added to striatal slices from fraction 12 and maintained through the experiment in the presence and absence of EPPTB. EPPTB (1 µmol/L) was added to striatal slices from fraction 3 and maintained throughout the experiment. (±)Methamphetamine-induced [^3^H]dopamine release was 12.42 ± 1.46 and this release was decreased to 4.20 ± 1.62 percent of content by 1 µmol/L EPPTB, *p* < 0.01; Student *t*-statistics for two-means, mean S.E.M., *n* = 4-8. (**B**) EPPTB (0.1 and 1 µmol/L) antagonized the effect of (±)methamphetamine on resting but not on electrical stimulation-induced [^3^H]dopamine release in rat striatum. For the experimental procedure, see Figure 2A,B. (±)Methamphetamine was added to striatal slices from fraction 8 in a concentration of 6.75 µmol/L. When used, EPPTB was added to the slices in a concentration of 0.1 or 1 µmol/L from fraction 1 and drugs were maintained throughout the experiment. Resting [^3^H]dopamine release (defined as the fractional release between fractions 8 and 17) was 0.37 ± 0.08 in control and 4.27 ± 0.46 percent of content in the presence of (±)methamphetamine, this release was decreased by EPPTB. The electrical-induced [^3^H]dopamine release (S2/S1) was 0.89 ± 0.03 in control and 1.86 ± 0.32 in the presence of (±)methamphetamine; this release was not altered by EPPTB. One-way ANOVA followed by the Dunnett’s test, F(3,26) = 13.990, *p* < 0.001 for resting [^3^H]dopamine release and F(3,26) = 5.554, *p* < 0.004 for electrical stimulation-induced [^3^H]dopamine release, # *p* < 0.05 control vs. (±)methamphetamine effect, * *p* < 0.05 (±)methamphetamine vs. (±)methamphetamine and EPPTB effect, mean ± S.E.M., *n* = 6-8.

**Figure 12 ijms-23-08543-f012:**
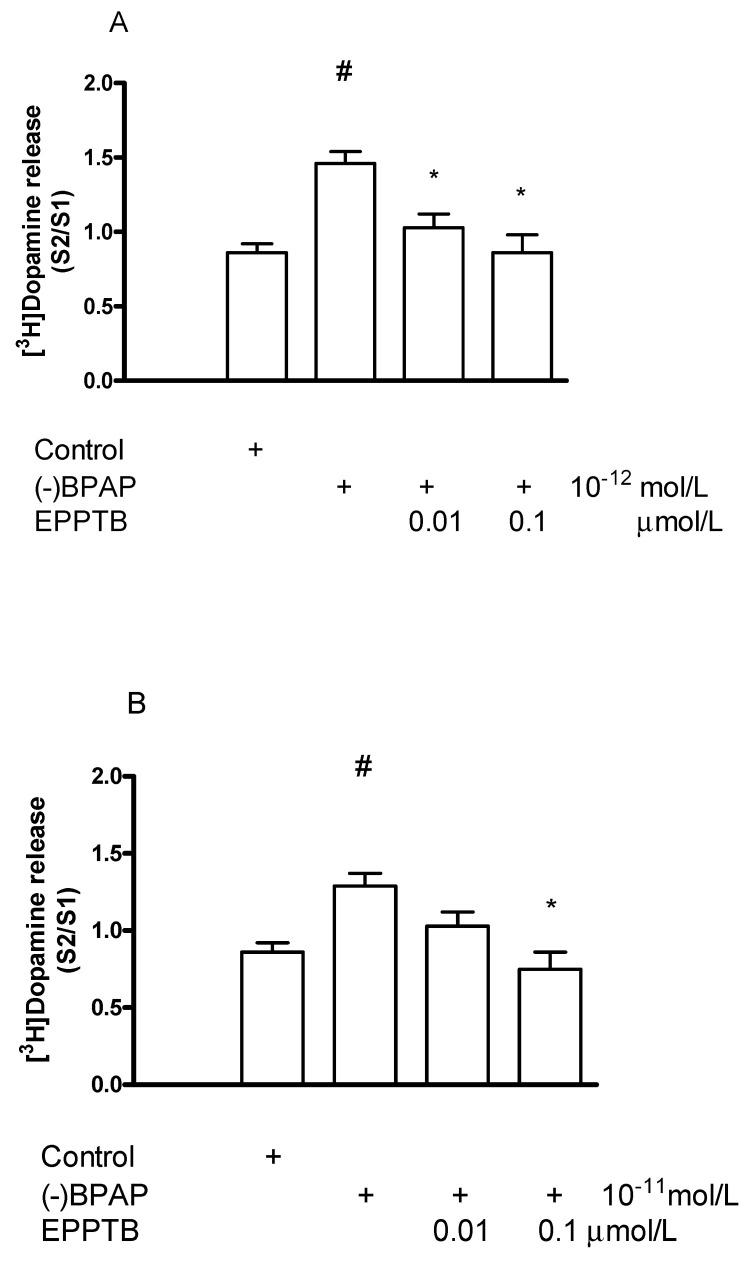
EPPTB reversed the (−)BPAP-induced [^3^H]dopamine release in rat striatum. For the experimental procedure, see Figure 2A. (−)BPAP was added to striatal slices from fraction 8 in a concentration of either 10^−12^ (**A**) or 10^−11^ mol/L (**B**) in the presence and absence of EPPTB. When used, EPPTB was added to striatal slices from fraction 1 in a concentration of either 0.01 or 0.1 µmol/L and drugs were maintained through the experiment. (**A**) One-way ANOVA followed by the Dunnett’s test, F(3,28) = 12.310, *p* < 0.001, # *p* < 0.01; Student *t*-statistics for two-means, control vs. (−)BPAP effect *p* < 0.05, (−)BPAP vs. (−)BPAP, and EPPTB (0.01 and 0.1 µmol/L) effects * *p* < 0.01, mean ± S.E.M., *n* = 7-10. (**B**) One-way ANOVA followed by the Dunnett’s test, F(3,23) = 3.890, *p* = 0.022, Student *t*-statistics for two-means, control vs. (−)BPAP effect # *p* < 0.01, (−)BPAP vs. (−)BPAP and EPPTB (0.1 µmol/L) effect * *p* < 0.01, mean ± S.E.M., *n* = 6-8.

**Figure 13 ijms-23-08543-f013:**
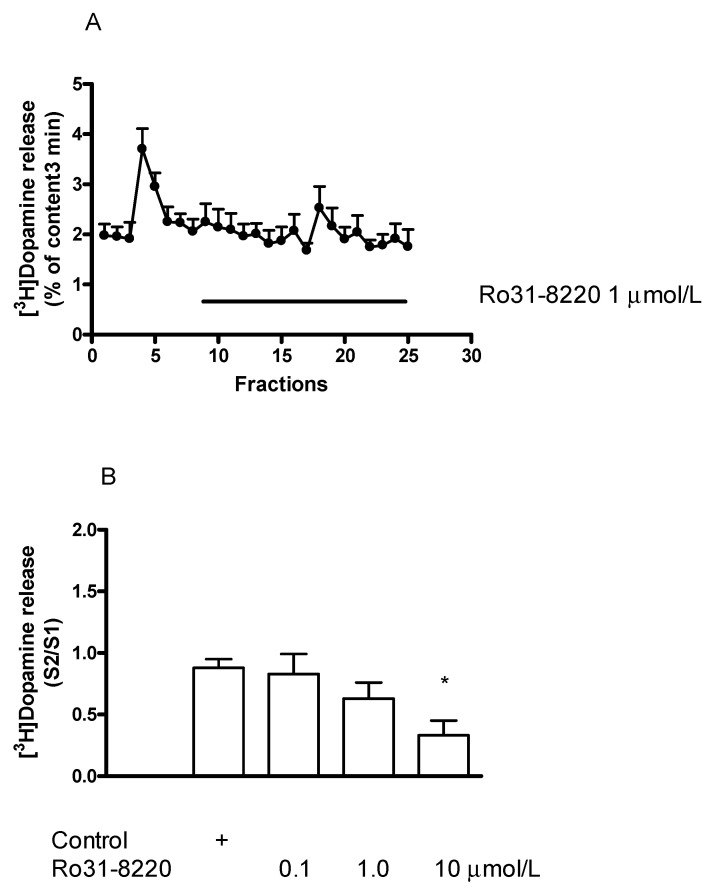
Effect of Ro31-8220 on [^3^H]dopamine release from rat striatum. For the experimental procedure, see Figure 2A. (**A**) Filled circles indicate [^3^H]dopamine release in the presence and absence of Ro31-8220. Ro31-8220 was added to striatal slices from fraction 8 in a concentration of 1 µmol/L and maintained throughout the experiment. Resting release expressed by the B2/B1 ratio was 0.81 ± 0.03 in control and it was 0.90 ± 0.09 in the presence of Ro31-8220, the difference was not significant. The electrical stimulation-induced release expressed by the S2/S1 ratio was 0.80 ± 0.07 in control and it was 0.47 ± 0.10 in the presence of Ro31-8220, *p* < 0.05. Student *t*-statistics for two-means, mean ± S.E.M., *n* = 3-4. (**B**) Ro31-8220 concentration-dependently decreased electrical stimulation-induced [^3^H]dopamine release from rat striatum. Ro31-8220 was added to striatal slices from fraction 8 in concentrations varied from 0.1 to 10 µmol/L and maintained through the experiment. 10 µmol/L Ro31-8220 reduced electrical stimulation-induced [^3^H]dopamine release, * *p* < 0.01. One-way ANOVA followed by the Dunnett’s test, F(3,13) = 4.489, *p* = 0.022, mean ± S.E.M., *n* = 3-6.

**Figure 14 ijms-23-08543-f014:**
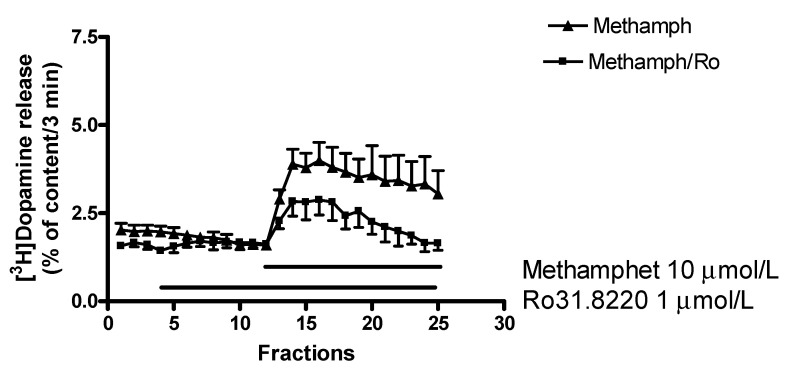
Ro31-8220 decreased the (±)methamphetamine (methamph)-induced non-vesicular. [^3^H]dopamine release in rat striatum. For the experimental procedure, see Figure 2B. (±)Methamphetamine was added to striatal slices from fraction 12 in a concentration of 10 μmol/L in the presence and absence of Ro31-8220. Ro31-8220 (1 µmol/L) was added to striatal slices from fraction 4 and drugs were maintained through the experiment. (±)Methamphetamine-induced non-vesicular [^3^H]dopamine release was 13.31 ± 2.67 and this release was decreased to 4.57 ± 1.48 percent of content by 1 µmol/L Ro31-8220, *p* < 0.05. Student *t*-statistics for two-means, mean ± S.E.M., *n* = 4-4.

**Figure 15 ijms-23-08543-f015:**
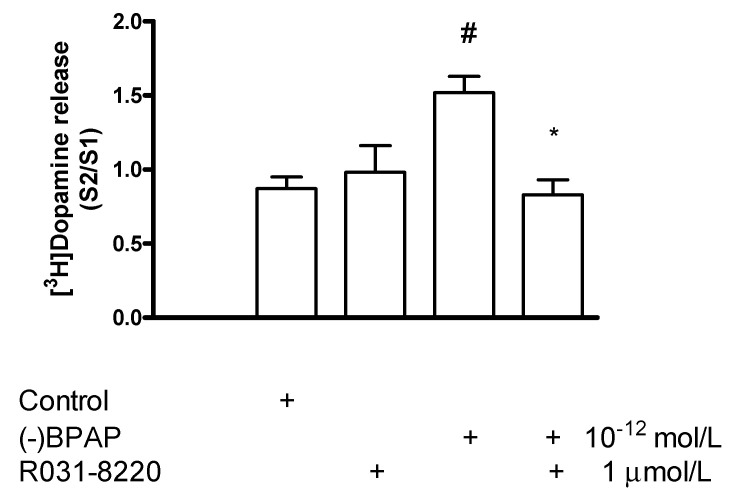
Ro31-8220 reversed (−)BPAP-induced [^3^H]dopamine release in rat striatum. For the experimental procedure, see Figure 2A. (−)BPAP was added to striatal slices from fraction 8 in a concentration of 10^−12^ mol/L in the presence and absence of Ro31-8220. When used, Ro31-8220 was added to striatal slices from fraction 1 in a concentration of 1 µmol/L and drugs were maintained throughout the experiment. One-way ANOVA followed by the Dunnett’s test, F(3,21) = 9.235, *p* < 0.001, control vs. (−)BPAP effect # *p* < 0.05. Student *t*-statistics for two-means, (−)BPAP vs. (−)BPAP plus Ro31-8220 effect * *p* = 0.001, mean ± S.E.M., *n* = 3-8.

**Figure 16 ijms-23-08543-f016:**
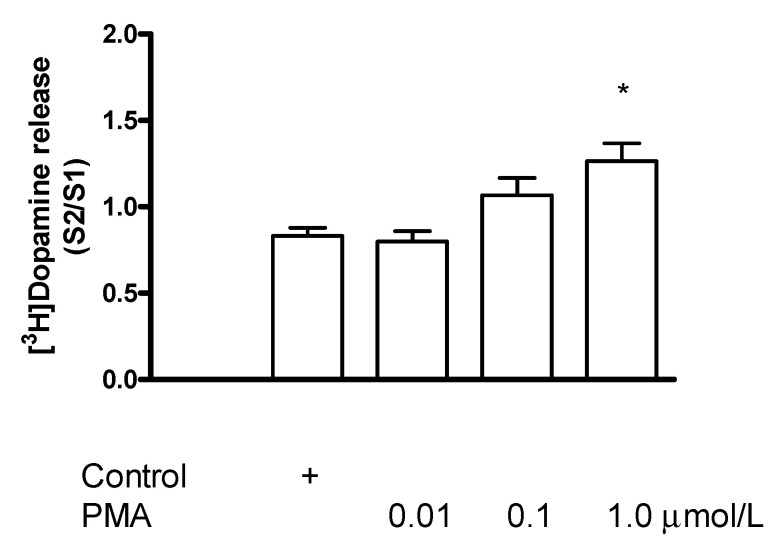
Phorbol 12-myristate 13-acetate (PMA) increased electrical stimulation-induced [^3^H]dopamine release from rat striatum. For the experimental procedure, see Figure 2A. PMA was added to striatal slices from fraction 8 in a concentration range varied from 0.01 to 1 µmol/L and maintained through the experiment. Resting release expressed by the B2/B1 ratio was 0.79 ± 0.03 in control and it was 0.71 ± 0.23 in the presence of 1 µmol/L PMA, not differing significantly. 1 µmol/L PMA increased the electrical stimulation-induced [^3^H]dopamine release, * *p* < 0.01. One-way ANOVA followed by the Dunnett’s test, F(3,12) = 7.196, *p* = 0.0051, * *p* < 0.01, mean ± S.E.M., *n* = 4.

**Figure 17 ijms-23-08543-f017:**
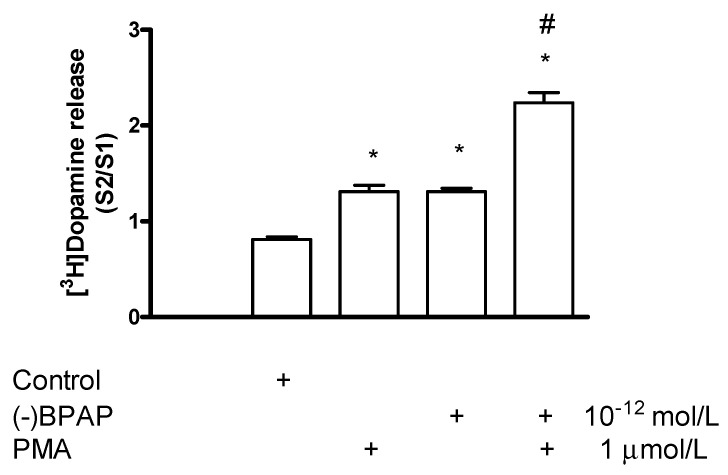
Phorbol 12-myristate 13-acetate (PMA) potentiated the (−)BPAP-stimulated [^3^H]dopamine release in rat striatum. For the experimental procedure, see Figure 2A. (−)BPAP was added to striatal slices from fraction 8 in a concentration of 10^−12^ mol/L. PMA was added to striatal slices from fraction 12 at a concentration of 1 µmol/L, and drugs were maintained throughout the experiment. One-way ANOVA followed by the Dunnet’s test, F(3,28) = 82.75, *p* < 0.0001, * *p* < 0.05. Student *t*-statistics for two-means, PMA vs. PMA plus (−)BPAP effect # *p* < 0.0001, mean ± S.E.M., *n* = 8.

**Figure 18 ijms-23-08543-f018:**
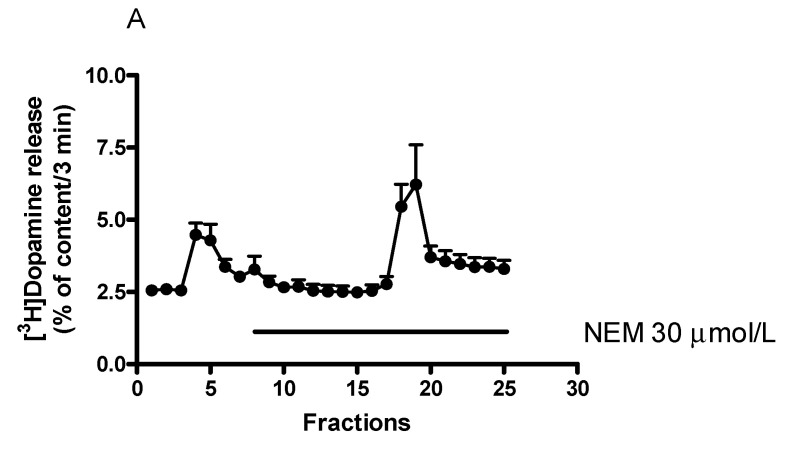
Effect of N-ethylmaleimide (NEM) on [^3^H]dopamine release from rat striatum. For the experimental procedure, see Figure 2A. (**A**) Filled circles indicate [^3^H]dopamine release in the presence and absence of NEM. NEM was added to striatal slices from fraction 8 in a concentration of 30 µmol/L and maintained through the experiment. Resting release expressed by the B2/B1 ratio was 0.93 ± 0.06 in control, and it was 1.07 ± 0.07 in the presence of NEM, not differing significantly. The electrical stimulation-induced release expressed by the S2/S1 ratio was 0.88 ± 0.06 in the control and 1.63 ± 0.23 in the presence of 30 µmol/L NEM, *p* = 0.009. Student *t*-statistics for two-means, mean ± S.E.M., *n* = 7-7. (**B**) NEM increased electrical stimulation-induced [^3^H]dopamine release in rat striatum. NEM was added to striatal slices from fraction 8 in a concentration range varied from 10 to 100 µmol/L and maintained through the experiment. One-way ANOVA followed by the Dunnett’s test, F(3,11) = 13.32, *p* = 0.006, * *p* < 0.05, mean ± S.E.M., *n* = 3-4.

**Figure 19 ijms-23-08543-f019:**
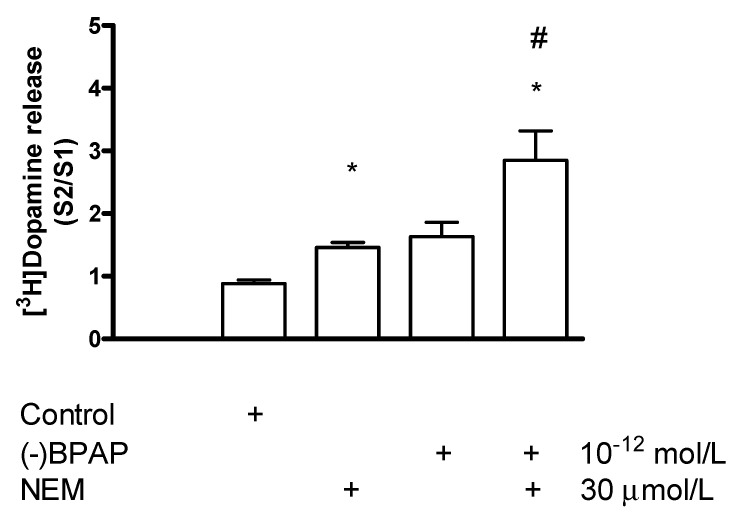
N-Ethylmaleimide (NEM) NEM potentiated the (−)BPAP-stimulated [^3^H]dopamine release in rat striatum. For the experimental procedure, see Figure 2A. (−)BPAP was added to the slices from fraction 8 in a concentration of 10^−12^ mol/L. NEM was added to the slices from fraction 12 in a concentration of 30 µmol/L and drugs were maintained throughout the experiment. One-way ANOVA followed by the Dunnett’s test, F(3,26) = 8.740, *p* = 0.048, control vs. NEM and (−)BPAP effect # *p* < 0.05. Student *t*-statistics for two-means, control vs. NEM effect * *p* < 0.01, NEM vs. NEM and (−)BPAP effect * *p* < 0.05, (−)BPAP vs. (−)BPAP/NEM effect * *p* < 0.05, mean ± S.E.M., *n* = 7-8.

**Figure 20 ijms-23-08543-f020:**
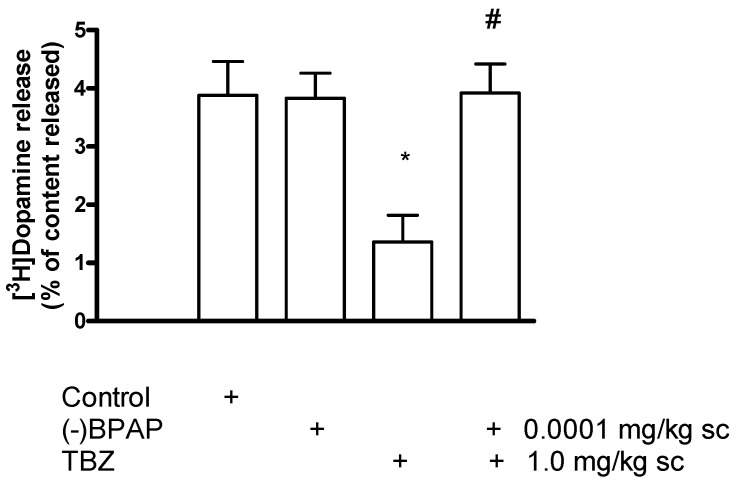
(−)BPAP reversed tetrabenazine (TBZ)-induced inhibition of [^3^H]dopamine release from rat striatum. Rats were injected with saline, TBZ (1 mg/kg), (−)BPAP (0.0001 mg/kg), and TBZ (1 mg/kg) plus (−)BPAP (0.0001 mg/kg) sc 60 min before the experiments. Striatal slices were prepared, loaded with [^3^H]dopamine and superfused. The resting and the electrical stimulation (40 V, 10 Hz, 2-msec for 3 min in fraction 4)-induced [^3^H]dopamine release was determined. Electrical stimulation markedly increased the release of [^3^H]dopamine from striatal slices obtained from saline-treated rats, this release was inhibited by injection of TBZ. (−)BPAP reversed the inhibitory effect of TBZ on [^3^H]dopamine release in the striatum of rats concomitantly treated with the two drugs. (−)BPAP treatment did not alter electrical stimulation-induced [^3^H]dopamine release. One-way ANOVA followed by the Dunnett’s test, F(3,26) = 8.617, *p* = 0.0004, saline vs. TBZ pretreated groups * *p* < 0.01. Student *t*-statistics for two means, TBZ vs. TBZ and (−)BPAP effect # *p* < 0.001, mean ± S.E.M., *n* = 7-8.

**Figure 21 ijms-23-08543-f021:**
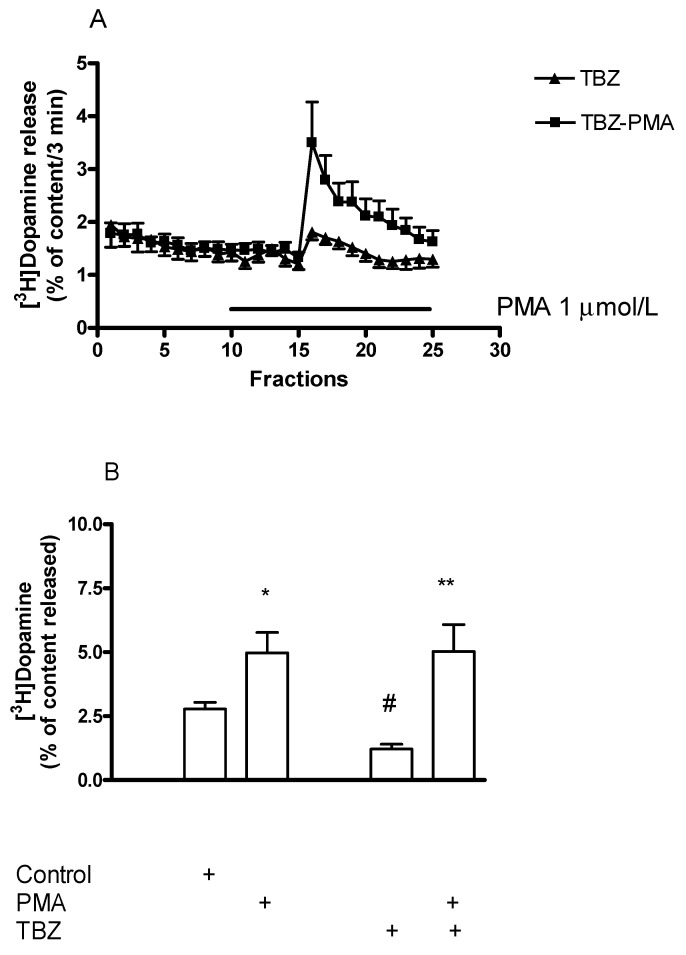
Phorbol 12-myristate 13-acetate (PMA) reversed the inhibitory effect of tetrabenazine (TBZ) on [^3^H]dopamine release in rat striatum. Striatal slices were prepared from non-treated and TBZ-pretreated (1 mg/kg sc 60 min before the experiment) rats, loaded with [^3^H]dopamine and superfused. The resting and the electrical stimulation (40 V, 10 Hz, 2-msec for 3 min in fraction 16)-induced [^3^H]dopamine release was determined as a fractional rate. (**A**) PMA was added from fraction 10 in a concentration of 1 µmol/L to striatal slices obtained from TBZ-pretreated rats and was maintained throughout the experiment. (**B**) TBZ pretreatment decreased the electrical stimulation-induced [^3^H]dopamine release from striatum: the release was 2.78 ± 0.19 in control and 1.21 ± 0.19 percent of content in tetrabenazine-pretreated rat striatum, # *p* < 0.05. PMA (1 µmol/L) increased the electrical stimulation-induced [^3^H]dopamine release in non-treated and TBZ-pretreated rat striatum by 178 and 475%, respectively. Student *t*-statistics for two-means, * *p* < 0.05, ** *p* < 0.01, mean ± S.E.M., *n* = 6-8.

**Figure 22 ijms-23-08543-f022:**
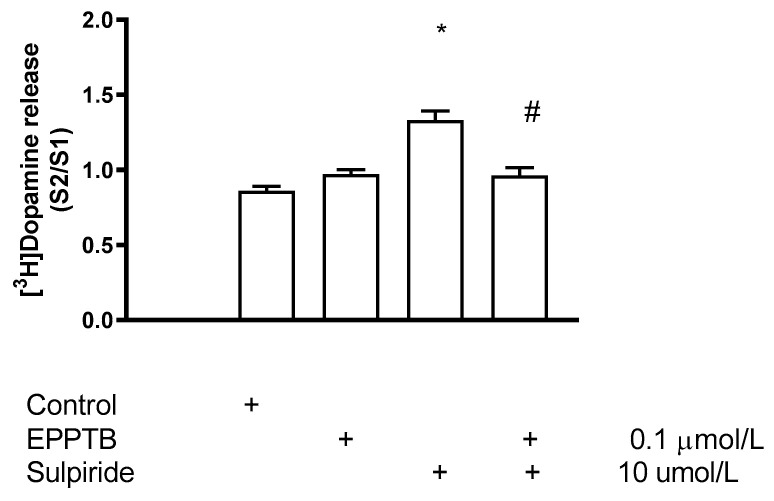
EPPTB reversed the sulpiride-induced increase of [^3^H]dopamine release from rat striatum. For the experimental procedure, see Figure 2A. EPPTB or sulpiride was added to the slices from fraction 8 in a concentration of 0.1 or 10 µmol/L, respectively. When EPPTB and sulpiride were added in combination, EPPTB (0.1 µmol/L) was added from fraction 1 and sulpiride (10 µmol/L) was added from fraction 8 and drugs were maintained throughout the experiment. Sulpiride did not alter resting [^3^H]dopamine release on its own. One-way ANOVA followed by the Dunnett’s test, F(3,21) = 21.65, *p* = 0.001, control vs. sulpiride treated groups * *p* < 0.05. Student *t*-statistics for two means, sulpiride vs. sulpiride and EPPTB effect # *p* < 0.05, mean ± S.E.M., *n* = 4-7.

**Figure 23 ijms-23-08543-f023:**
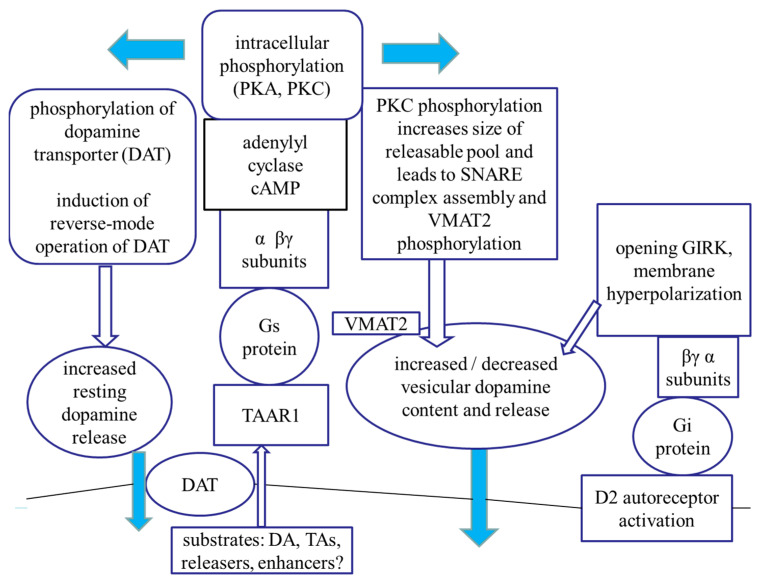
Hypothetical model of the trace amine-associated receptor 1 (TAAR1) signaling involved in the regulation of presynaptic dopaminergic neurotransmission. Activation of the Gs protein-coupled TAAR1 by classical or trace amines and exogenous releaser drugs (amphetamines) results in adenylyl cyclase activation followed by an increased intracellular cAMP production and, as a consequence, an increase in PKA/PKC-mediated intracellular phosphorylation. Activation of PKC phosphorylates plasma membrane dopamine transporter (DAT) as well as proteins involved in SNARE core complex assembly leading to non-vesicular and vesicular dopamine release, respectively. Moreover, PKC also phosphorylates vesicular monoamine transporter 2 (VMAT2) and the increased dopamine accumulation causes elevated vesicular dopamine content and release. Dopamine released into the synaptic cleft activates D2 autoreceptor-mediated presynaptic feedback inhibition, which then leads to the opening of GIRK and the resulted hyperpolarization of the cell membrane suspends further release of dopamine. Our working hypothesis is that TAAR1 possesses a central role in the regulation of DAT, dopamine-containing vesicle operation, readily releasable pool and SNARE core complex assembly and feedback inhibition of dopamine release also. We have speculated that the enhancer drugs ((−)BPAP) activate TAAR1 signaling and the reserpine-sensitive PKC pool and phosphorylation of the proteins in SNARE core complex and VMAT2 leads to increased vesicular dopamine release, i.e., enhancer effect.

## Data Availability

Archived laboratory documents.

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
