# Peer review of "Enhancer Regulation of Dopaminergic Neurochemical Transmission in the Striatum"

_ijms, 2022, doi:10.3390/ijms23158543_

Round 1
Reviewer 1 Report
The first part of the introduction is similar to an abstract. Then, the second part is similar to a review, not an introduction meant for an Article. I strongly suggest revising the structure of the introduction.
The authors should mention how the TAAR family has a crucial role in characterizing the neurobiology of the central nervous system.
The introduction lacks completely the evidence/motivation for the link between TAARs and CAE.
At the end of each paragraph, despite the detailed description of the evidence so far, a question should arise as the aim of the research. What is not clear yet? Why do we need to know that?
As so, in the conclusion paragraph. What is the impact of the study?
The study design and methods are detailed and well-described.
Author Response
Reviewer 1
We appreciate the Reviewer’s effort for his/her constructive comment.
The first part of the introduction is similar to an abstract. Then, the second part is similar to a review, not an introduction meant for an Article. I strongly suggest revising the structure of the introduction.
1) The structure of the Introduction was revised.
The authors should mention how the TAAR family has a crucial role in characterizing the neurobiology of the central nervous system.
2) The wide distribution of TAAR in the central nervous system is discussed in the Introduction showing its role in neurobiology of the CNS.
The introduction lacks completely the evidence/motivation for the link between TAARs and CAE.
3) Possible link between TAAR and CAE was added into the Introduction as an aim for the current investigation.
This possible link between TAAR and CAE is further discussed in section Discussion 4.1.1.
At the end of each paragraph, despite the detailed description of the evidence so far, a question should arise as the aim of the research. What is not clear yet? Why do we need to know that?
4) We have focus to make the aim of this research more clearly explained in the last paragraph of the Introduction.
As so, in the conclusion paragraph. What is the impact of the study?
The study design and methods are detailed and well-described.
5) CAE substances as possible therapeutics are hypothesized and added in the Conclusion.

Reviewer 2 Report
This is an interesting study about the regulation of dopaminergic neurochemical transmission in the striatum where the authors have performed several experiments in an attempt to elucidate the role of (-)BPAP compound on this complex neurochemical system. However, I have several comments that I'd like to see clarified.
Introduction:
Line 52: “The precise mechanism behind the enhancer activity is not yet fully clarified.” The authors should provide proper bibliographic references that confirm this statement.
In addition to the brain TAAR1 is expressed in other organs? And within the brain there is any difference on the expression across different region? E.g. nucleon accumbens and ventral tegmentar area? This should be referred on introduction.
Figure 2, which shows the chemical structures of the trace amines it doesn't seem to add much to the work, maybe it should be submitted as a supplementary figure. Furthermore, if the authors' objective is to mention that they are very similar to biogenic amines, it is important that this image includes these same biogenic amines so that the reader can compare them. Or perhaps, instead of images 1, 2, 3 and 4, it might be more interesting to have just one figure with the chemical structures of all compounds.
Authors should review the way they cite bibliographic references, there are many statements without any kind of reference. For example: line 37, 52, 64, 71, 73, 105, 163, etc..
Results:
Line 242, “is a result of exocytosis (Harsing, 2008).”. The reference is in wrong format.
Line 287, the representation of the data in the graphs of figure 6 is too small, which makes it impossible to evaluate the effect of each drug on its own. Authors should modify graphics, for example by changing the scale of the graph or using a color system. Why are standard deviations not shown on the graph?
From line 300 to line 308, the authors should revise this text. First they describe the effect in graph 7B and then in 7A and the because they need to add some data regarding the difference in ratios between control and methamphetamine (as they refer in the figure legend).
Line 309, if 10µmol/L have the strongest effect on dopamine release why on figure 7A add 6,75µmol/L and not 10µmol/L?
Line 310, on graphic 7B the conditions 6,75 and 10µmol/L are statically significant from the other conditions?
Line 324, why the authors used a concentration of compound (-)BPAP of 10-12 mol/L?
Line 329, On figure 8 if graphic A is the control condition and graphic B the presence of (-)BPAP, why they are not represented on the same graphic?
Line 519, if the authors tested different concentrations of Ro31-8220 (fig. 16B) why on figure 16A they chose 1 μmol/L? and not for example 10 μmol/L which is statistically significant?
In general, the graphics are too big with too much space between them, the way the information is being displayed should be revised to make it more harmonious and clearer. Additionally, the figure caption in many cases has information that would help to analyze and interpret the results. This information should be removed from the captions and incorporated into the text. An example of this is figure 6, 7, 8, 9, 10, 11, 13, 15, etc…
Discussion:
Line 732, “It was found that EPPTB, used in specific concentration,” what specific concentrations are?
Conclusion:
Line 1064, “The therapeutic aspects of the enhancer compound (-)BPAP, however, remain to be elucidated.”. What kind of therapeutic aspects could these be? It would be quite interesting for the authors to discuss the therapeutic potential of this compound in the discussion section.
Author Response
Reviewer 2
We appreciate the Reviewer’s effort for his/her constructive comment.
Introduction:
Line 52: “The precise mechanism behind the enhancer activity is not yet fully clarified.” The authors should provide proper bibliographic references that confirm this statement.
In addition to the brain TAAR1 is expressed in other organs? And within the brain there is any difference on the expression across different region? E.g. nucleon accumbens and ventral tegmentar area? This should be referred on introduction.
Figure 2, which shows the chemical structures of the trace amines it doesn't seem to add much to the work, maybe it should be submitted as a supplementary figure. Furthermore, if the authors' objective is to mention that they are very similar to biogenic amines, it is important that this image includes these same biogenic amines so that the reader can compare them. Or perhaps, instead of images 1, 2, 3 and 4, it might be more interesting to have just one figure with the chemical structures of all compounds.
Authors should review the way they cite bibliographic references, there are many statements without any kind of reference. For example: line 37, 52, 64, 71, 73, 105, 163, etc..
Line 52, Reference added.
TAAR family expression in other peripheral organs and brain areas is added into the Introduction.
Figs. 1, 2, 3 and 4 are combined as was suggested. Please, see Fig. 1.
Additional references were added in the Introduction as was suggested by the Reviewer.
Results:
Line 242, “is a result of exocytosis (Harsing, 2008).”. The reference is in wrong format.
Line 287, the representation of the data in the graphs of figure 6 is too small, which makes it impossible to evaluate the effect of each drug on its own. Authors should modify graphics, for example by changing the scale of the graph or using a color system. Why are standard deviations not shown on the graph?
From line 300 to line 308, the authors should revise this text. First they describe the effect in graph 7B and then in 7A and the because they need to add some data regarding the difference in ratios between control and methamphetamine (as they refer in the figure legend).
Line 309, if 10µmol/L have the strongest effect on dopamine release why on figure 7A add 6,75µmol/L and not 10µmol/L?
Line 310, on graphic 7B the conditions 6,75 and 10µmol/L are statically significant from the other conditions?
Line 324, why the authors used a concentration of compound (-)BPAP of 10-12 mol/L?
Line 329, On figure 8 if graphic A is the control condition and graphic B the presence of (-)BPAP, why they are not represented on the same graphic?
Line 519, if the authors tested different concentrations of Ro31-8220 (fig. 16B) why on figure 16A they chose 1 μmol/L? and not for example 10 μmol/L which is statistically significant?
In general, the graphics are too big with too much space between them, the way the information is being displayed should be revised to make it more harmonious and clearer. Additionally, the figure caption in many cases has information that would help to analyze and interpret the results. This information should be removed from the captions and incorporated into the text. An example of this is figure 6, 7, 8, 9, 10, 11, 13, 15, etc…
Line 242, corrected.
Line 287, figures are modified as suggested. The mean±S.E.M. is shown in the graphs. Added to legend: Control for Figs. 6A and B (in the revised version Fig.3A and B) is shown in Fig. 2B.
Lines from 300 to 308 are corrected. Figs. 7A and 7B (in the revised version Fig. 4A and B) and the legends are corrected.
Line 309. The releasing effect of (±)methamphetamine was investigated in both 6.75 μmol/L concentration (in the revised version Fig. 3A, Figs. 4A and B, Figs. 8A and B, Fig. 11B) and in 10 μmol/L concentration (Fig. 3A, Fig. 11A, Fig. 14).
Line 310. We have added the one-way-ANOVA followed by the Dunnet’s test statistical analysis in the legend of Fig 4A.
Line 324. The enhancer effect of (-)BPAP on [3H]dopamine release appeared in concentrations of 10-12 and 10-11 mol/l, both concentrations of (-)BPAP were further investigated in EPPTB drug combinations (Fig. 12A and B).
Line 329. We have combined release data in control (Fig. 5A) and in the presence of 10-12 mol/L concentration of (-)BPAP (Fig. 5B) in a single graph. The release curves, however, overlapped and the differences could not be clearly seen. Therefore, we would like to present these data in two separate graphs.
Line 519. The legend of Fig. 13A indicates that Ro31-8220, added in 1 μmol/L concentration, significantly decreased [3H]dopamine release, the used statistical analysis was Student t-statistics for two means. Therefore, this concentration was used in combination with (±)methamphetamine (Fig. 14) or (-)BPAP (Fig. 15). When a concentration-effect analysis was made for Ro31-8220 (concentration range was 0.1 to 10 μmol/L, one-way ANOVA followed by the Dunnett’s test indicated significant p value at 10 μmol/L concentration (Fig. 13B).
We have improved the figures as Reviewer suggested. We made all efforts to show results in figures for being self-explanatory, figure captions and the figure legends were prepared accordingly.
Discussion:
Line 732, “It was found that EPPTB, used in specific concentration,” what specific concentrations are?
We used EPPTB in a concentration range of 0.01 to 1.0 μmol/L when the pre se effect of the drug on [3H]dopamine release was investigated (Fig. 10B). These concentrations of EPPTB were applied in combination studies of (±)methamphetamine (Fig. 11A and B) or (-)BPAP (Fig. 12A and B). In our studies, we chose these concentrations of EPPTB taken from the papers of Bradaia et al., 2009 and Revel et al., 2011 (see References) as its specific concentrations for inhibition of TAAR1.
Conclusion:
Line 1064, “The therapeutic aspects of the enhancer compound (-)BPAP, however, remain to be elucidated.”. What kind of therapeutic aspects could these be? It would be quite interesting for the authors to discuss the therapeutic potential of this compound in the discussion section.
The potential clinical uses of the CAE compounds, including (-)BPAP are summarized in the Conclusion.
